



# Source Mechanisms and transport Patterns of tropospheric BrO: Findings from long-term MAX-DOAS Measurements at two Antarctic Stations

Udo Frieß[1], Karin Kreher[3], Richard Querel[2], Holger Schmithüsen[4], Dan Smale[2], Rolf Weller[4], and Ulrich Platt[1]

[1]Institute of Environmental Physics, Heidelberg University, Germany
[2]National Institute of Water and Atmospheric Research, Lauder, New Zealand
[3]BK Scientific GmbH, Mainz, Germany
[4]Alfred Wegener Institute for Polar and Marine Research, Bremerhaven, Germany

**Correspondence:** Udo Frieß (udo.friess@iup.uni-heidelberg.de)

**Abstract.** The presence of reactive bromine in Polar Regions is a widespread phenomenon that plays an important role in the photochemistry of the Arctic and Antarctic lower troposphere, including the destruction of ozone, the disturbance of radical cycles, and the oxidation of gaseous elemental mercury. The chemical mechanisms leading to the heterogeneous release of gaseous bromine compounds from saline surfaces are in principle well understood. There are, however, substantial uncertainties

about the contribution of different potential sources to the release of reactive bromine, such as sea ice, brine, aerosols and the snow surface, as well as about the seasonal and diurnal variation and the vertical distribution of reactive bromine. Here we use continuous long-term measurements of the vertical distribution of bromine monoxide (BrO) and aerosols at the two Antarctic sites Neumayer (NM) and Arrival Heights (AH), covering the periods of 2003 - 2021 and 2012 - 2021, respectively, to investigate how chemical and physical parameters affect the abundance of BrO. We find the strongest correlation between BrO

and aerosol extinction ($R = 0.56$ for NM and $R = 0.28$ for AH during spring), suggesting that the heterogeneous release of $Br_2$ from saline airborne particles (blowing snow and aerosols) is a dominant source for reactive bromine. Positive correlations between BrO and contact time of air masses both, with sea ice and the Antarctic ice sheet suggest that reactive bromine is not only emitted by the sea ice surface, but by the snowpack on the ice shelf and in the coastal regions of Antarctica. In addition, the open ocean appears to represent a source for reactive bromine during late summer and autumn when the sea ice extent is at

its minimum. A source-receptor analysis based on back trajectories together with sea ice maps shows that main source regions for BrO at NM is the Weddell Sea and the Filchner-Ronne Ice Shelf, as well as coastal polynias where sea ice is newly formed. A strong morning peak in BrO frequently occuring during summer, and particular during autumn, suggests a night-time built up of $Br_2$ by heterogeneous reaction of ozone on the saline snow pack in the vicinity of the measurement sites. We furthermore show that BrO can be sustained for several days while travelling across the Antarctic continent in the absence of any saline

surfaces that could serve as a source for reactive bromine.



## 1 Introduction

It is known since the 1980s that reactive bromine compounds play an important role in the troposphere of Polar Regions (Barrie et al., 1988). Bromine radicals ($Br_x$=[Br, BrO]) are known to have a strong impact on the oxidative capacity of the polar atmosphere already at mixing ratios in the lower parts per trillion (ppt) range. They cause photochemical destruction of ozone, which frequently drops to near-zero values in the presence of reactive bromine during so-called ozone depletion events (ODEs) (Hausmann and Platt, 1994; Helmig et al., 2012), but also affect other important photochemical species, such as $HO_x$ ($OH + HO_2$) and $NO_x$ ($NO + NO_2$), and volatile organic compounds (Barrie and Platt, 1997; Platt and Hönninger, 2003; von Glasow and Crutzen, 2007; Simpson et al., 2007; Abbatt et al., 2012; Simpson et al., 2015). Furthermore, bromine monoxide (BrO) is responsible for the oxidation of elemental mercury, and the conversion from its insoluble gaseous elementary form to soluble oxidised compounds, leading to an enhanced deposition of mercury species into the vulnerable polar ecosystems (Lu et al., 2001; Ariya et al., 2002; Ebinghaus et al., 2002; Steffen et al., 2008; Dommergue et al., 2010). Satellite measurements of BrO reveal that the presence of reactive bromine is a widespread phenomenon, covering several millions of square kilometres of the sea ice covered oceans and parts of the adjacent Antarctic and Arctic land masses during polar spring (Wagner and Platt, 1998; Richter et al., 1998; Schönhardt et al., 2012; Bougoudis et al., 2020).

Reactive bromine is released to the gas phase via heterogeneous reactions on saline surfaces with low pH, which contain a sufficient amount of bromide ($Br^-$), by the following reaction cycle (Tang and McConnell, 1996; Vogt et al., 1996; Wennberg, 1999):

$$Br_2 + h\nu \quad \rightarrow \quad 2Br \tag{R1}$$

$$Br + O_3 \quad \rightarrow \quad BrO + O_2 \tag{R2}$$

$$BrO + HO_2 \quad \rightarrow \quad HOBr + O_2 \tag{R3}$$

$$HOBr + Br^-_{(aq)} + H^+_{(aq)} \quad \rightarrow \quad Br_2 + H_2O \tag{R4}$$

The fact that each gaseous HOBr molecule reacting on a saline surface via R4 yields up to two bromine atoms in the gas phase after photolysis of $Br_2$ (R1) can lead to an exponential increase of the amount of gaseous reactive bromine, the so-called bromine explosion (Platt and Hönninger, 2003). Apart from this bromine explosion mechanism, which requires sunlight, bromine can also be released during the night by reaction of ozone on saline surfaces via the following reactions (Oum et al., 1998; Artiglia et al., 2017; Pratt et al., 2013):

$$O_3 + Br^-_{(aq)} \quad \rightarrow \quad BrO^- + O_2 \tag{R5}$$

$$BrO^-_{(aq)} + H^+_{(aq)} \quad \rightarrow \quad HOBr_{(aq)} \tag{R6}$$

The HOBr produced by R6 leads to a release of molecular bromine ($Br_2$) to the gas phase via R4. In contrast to the HOBr-driven cycle (R1 - R4), the ozone-driven bromine release initiated by R5 and R6 is not self-accelerating, but can lead to the accumulation of large amounts of molecular bromine in the nocturnal polar boundary layer, and its rapid photolysis during sunrise via R1 can lead to a peak in the abundance of reactive bromine in the morning (Simpson et al., 2018). Apart from



the aforementioned heterogeneous mechanisms, OH-mediated release of reactive bromine represents a further possible release process that also proceeds in the absence of sunlight (Halfacre et al., 2019).

The release mechanisms for reactive bromine are thought to be well understood, and photochemical simulations reaching from one-dimensional models (Piot and von Glasow, 2008; Toyota et al., 2014b, a) to full 3D CTMs (Toyota et al., 2011; Herrmann et al., 2021, 2022) are nowadays capable of reproducing surface- and satellite borne observations of ozone and BrO levels in the polar atmosphere. There is, however, still a large uncertainty in the contribution of different surface types and release mechanisms to the reactive bromine budget (Abbatt et al., 2012). A brine layer forming on top of the sea ice, in

particular during its formation in open leads, is thought to be a major source of reactive bromine (Peterson et al., 2016), and frost flowers forming on top of the brine layer might provide additional surface areas for heterogeneous bromine release (Kaleschke et al., 2004), although their total area is now thought to be too small to have a strong impact on the total reactive bromine budget. Saline particles can become airborne at high wind speeds, e.g. by dispersion, saltation processes, wind pumping and by blowing snow impacting brine surfaces (Morin et al., 2008). Indeed, surface observations show high amounts of BrO in

the presence of blowing snow (Frieß et al., 2011; Peterson et al., 2017), suggesting that these serve as a source for reactive bromine. Airborne snow particles and aerosols with high salinity provide a transport pathway for bromine compounds to the coastal inland, where their deposition can lead to an accumulation of halides on snow surfaces, from which reactive bromine can be released at a later time (Piot and von Glasow, 2008).

The abundance of reactive bromine is expected to be controlled by a multitude of dynamical and chemical parameters, such

as the contact of air masses with sea ice (Frieß et al., 2004; Jones et al., 2006; Bognar et al., 2020), wind speed, blowing snow and aerosols (Jones et al., 2009; Frieß et al., 2011), the availability of ozone (Helmig et al., 2012), ambient temperature (Pöhler et al., 2010), as well as atmospheric stability and solar radiation. Highest amounts of BrO are observed during polar spring, when the sea ice extent is at its maximum and the bromine release mechanism can proceed due to the availability of sunlight after polar sunrise.

Ground-based measurements of tropospheric BrO in Antarctica are very sparse and restricted to a few locations, (Frieß et al., 2004; Schofield et al., 2006; Prados-Roman et al., 2018; Nasse, 2019) and in many cases only performed during a short period of time (Kreher et al., 1997; Saiz-Lopez et al., 2007; Wagner et al., 2007; Hay, 2010; Roscoe et al., 2014; Zielcke, 2015). Here we investigate the sources for reactive bromine, as well as the impact of a variety of parameters, on the abundance of BrO and aerosols on the basis of continuous long-term Multi-Axis Differential Optical Absorption Spectroscopy (MAX-DOAS)

measurements at the two coastal sites Neumayer and Arrival Heights, which are located at adjacent sides of the Antarctic continent in the Atlantic and Pacific sector, respectively. Covering the period of 2003 to 2021 and 2012 to 2021, respectively, without any major gaps except for polar night, the measurements yield an unprecedented continuous data set, consisting of more than 100 000 pairs of BrO and aerosol vertical profiles in the lower troposphere. Together with co-located measurements meteorological and chemical parameters, as well as back trajectory simulations and sea ice maps, these offers the opportunity

to study the sources of reactive bromine and its impact on the chemical processes in the Antarctic troposphere.

This paper is structured as follows: Instruments and data analysis are presented in Section 2. This includes a description of the measurement sites (Section 2.1) as well as a technical description of both instruments (Section 2.2), the data analysis



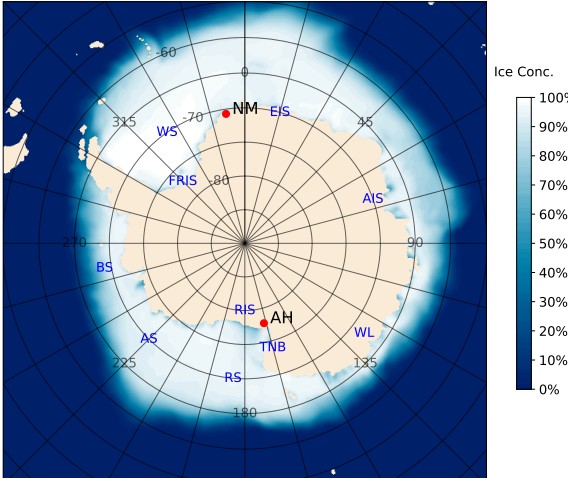

**Figure 1.** Location of AH and NM on the Antarctic continent (red dots). Blue to white colours indicate the mean (2003-2021) OSI-SAF sea ice concentration (see Section 2.6) from August to October. Land masses and shelf ice areas are shown in beige. The abbreviations for the geographical names shown in blue are listed in Table 1

(Sections 2.4 and 2.5), and a description of the back trajectory modelling that represents the basis for a BrO source-receptor analysis (Section 2.6), as well as the methodology for a statistical analysis of the observation (Section 2.7). Case studies illustrating the relationship between the vertical distribution of BrO, aerosols, atmospheric parameters, and the origin of air masses, are presented in Section 3.1. Seasonal and diurnal variation, as well as the vertical distribution of BrO are discussed in Section 3.2. The relationship of BrO and aerosols with meteorological and photochemical observables and with air mass history are presented in Section 3.3. Source regions for reactive bromine are identified on the basis of a source-receptor analysis presented in Section 3.4. Finally, Section 3.5 presents a case study for transcontinental transport of BrO from the Weddell sea over the Antarctic ice shield to the Ross Sea, showing that BrO can be sustained for several days in the absence of any source for reactive bromine.

## 2 Instruments and data analysis

### 2.1 Measurement locations

Long-term scattered sunlight measurements of BrO and the oxygen collision complex $O_4$ using MAX-DOAS instruments have been performed at the Antarctic research station Neumayer and at the Arrival Heights atmospheric observatory. The locations of the two measurement sites are depicted in Figure 1. The German Antarctic research station Neumayer (NM) is operated by the Alfred Wegener Institute, Helmholtz Centre for Polar and Marine Research (AWI), Germany. The station is located at $70°40'$S, $8°16'$W on the Ekström Ice Shelf (EIS) in front of the Queen Maud Land (QML), in a distance of about 5 km from



**Table 1.** List of abbreviations for the geographical names shown in Figure 1.

| Abbreviation | Site |
| --- | --- |
| FRIS | Filchner-Ronne Ice Shelf |
| EIS | Ekström Ice Shelf |
| AIS | Amery Ice Shelf |
| RIS | Ross Ice Shelf |
| TNB | Terra Nova Bay |
| WL | Wilkes Land |
| WS | Weddell Sea |
| RS | Ross Sea |
| AS | Amundsen Sea |
| BS | Bellingshausen Sea |

a naturally formed bay in the north-west of the station named Atka Bay, and approximately 15 km to the open ocean in the
north. The winterly built-up of sea ice usually starts in early May, and a large-scale sea ice cover persists until about the end of
November. The MAX-DOAS instrument is located on the roof of the Neumayer Air Chemistry Observatory at an altitude of
approximately 5 m above the snow surface.

Due to the flat surroundings of Neumayer Station, air masses can freely flow without any disturbing topographic obstacles.
Katabatic winds, i.e. gravitationally driven flow of cold and dense air down the slope of the Antarctic continent, are very
rare at Neumayer Station. The site is dominated by easterly winds which are associated with eastward moving cyclones, with
some rare occurrences of southerly to south-westerly winds (König-Langlo et al., 1997). The origin of the air masses will be
investigated in more detail in the framework of the trajectory modelling analysis as detailed in Section 2.6.

The Arrival Heights observatory (AH), which is part of New Zealand's Scott Base, is located on the opposite side of the
Antarctic continent at $77°49'$S, $166°39'$E at an altitude of 184 m asl on Hut Point Peninsula, which is part of Ross Island. The
Ross Ice Shelf (RIS) is located south of Arrival Heights, and the MAX-DOAS instrument points northwards over the Ross Sea
(RS), which is located at a distance of only 300 metre from the AH observatory. The RS is usually covered by sea ice from
late March to December.

Compared to NM, the topography around AH is far more complex, with the 3794 m high Mount Erebus to the East and the
Transantarctic Mountains to the West, leading to a more complex local meteorology with varying wind directions (Sinclair,
1988). The larger slope of the surrounding terrain compared to Neumayer, as well as the narrow flow channel formed by Ross
Island and the Transantarctic Mountains, lead to the frequent occurrence of strong katabatic winds from the interior of the
continent, observed as easterly and south-easterly winds due to deflection of the air masses by the mountains of Ross Island
(Godfrey and Clarkson, 1998; Seefeldt et al., 2003; van Lipzig et al., 2004).





## 2.2 MAX-DOAS Instruments

The spectrometer units of the MAX-DOAS instruments at NM and AH are of identical design, and technical details are already described elsewhere (Frieß et al., 2001, 2004, 2005; Frieß et al., 2010). In brief, they consist of two separate spectrometer units for the UV (320 - 420 nm, resolution 0.5 nm FWHM) and Vis (400 - 650 nm, 1.8 nm FWHM). Light is dispersed using holographic imaging gratings and detected using thermoelectrically cooled photo diode arrays (Hamamatsu ST3904-1024) with 1024 channels. Scattered sunlight is collected by telescope units and fed to the spectrometer units using quartz fibre

bundles.

The NM instrument has been in continuous operation since 1999. Initially, only zenith-sky measurements were performed until a custom-built MAX-DOAS telescope unit was installed in January 2003. The telescope unit is equipped with two rotatable prisms for UV and Vis, respectively, allowing to collect light from any elevation angle. The telescope points to the due North. The elevation angle sequence initially applied for the measurements at NM was $90°$ (zenith), $20°$, $10°$, $5°$, and $2°$.

In August 2006, a downward looking angle of $-5°$ was added in order to increase the sensitivity for trace gases below the instrument and within the snowpack (Frieß et al., 2010), and the elevation angles $1°$ as well as $-20°$ were finally added in September 2007.

The AH instrument has been in operation since August 1998. The entrance optics consists of a telescope housing located inside the AH observatory building. It is illuminated by scattered sunlight reflected from a solar tracker system installed on

top of the roof, consisting of two motor driven aluminium coated mirrors. The solar tracker is protected by an acrylic glass dome and allows for pointing to any direction in the sky. Various different viewing angle sequences were applied until October 2012, but none of them included a sufficient number of elevation angles for a retrieval of aerosols and trace gases. Since then, the instrument points to a fixed north-westerly azimuth direction ($305°$), and a measurement sequence consists of the elevation angles of $90°$, $20°$, $10°$, $5°$, $2°$, $1°$, $0°$ and $-4°$. The negative elevation angle, pointing downwards on the sea ice, is

of particular importance for retrieving information on the atmosphere below the instrument at AH due to its elevated location at 184 m altitude ASL. Unfortunately, the design of the solar tracker does not allow to include further downward looking directions that would further increase the information content of the air below the instrument. MAX-DOAS measurements are only performed for SZA < $89°$, i.e. from 1. August to 12. May at NM, and from 23. August to 22. April at AH.

Both instruments measure spectra with a total integration time of 3 minutes during the day when the solar zenith angle

(SZA) is below $90°$. The telescopes are equipped with mercury and halogen light sources, and calibration spectra (mercury and halogen lamp, dark current and offset) are recorded automatically each night.

## 2.3 Supporting Data

In the framework of this study, a variety of supporting data was used for the interpretation of the MAX-DOAS observations at both Antarctic sites. Hourly averaged meteorological data including surface temperature, wind speed and direction, as well as

relative humidity and pressure, has been measured using a NIWA automatic weather station at AH (https://cliflo.niwa.co.nz/, date of last access: 31.3.2022), and continuously performed meteorological observations at NM (https://www.pangaea.de/, date





of last access: 25.2.2022). Surface Ozone has been measured at AH by NIWA in collaboration with NOAA using a Thermo Scientific Model 49C (McClure-Begley et al., 2013, date of last access: 24.2.2022), and at NM by an Anysco Model O341M ozone analyser (https://www.pangaea.de/, date of last access: 25.2.2022). Vertical profiles of pressure, relative humidity and
temperature were taken from regularly performed radiosonde measurements at both sites using Vaisala RS41-SGP devices. At AH, these are provided by the Antarctic Meteorological Research Center (AMRC), launched at McMurdo station in the vicinity of AH (https://amrc.ssec.wisc.edu, date of last access: 24.2.2022). The radiosondes launched at NM also have an ozone sensor attached at least weekly, and multiple times per week during the ozone hole period in austral spring (Schmithüsen, 2022, date of last access: 2.5.2022). Furthermore, back trajectories are modelled based on meteorological fields from the Global
Data Assimilation System (GDAS) NCEP GDAS/FNL 0.25 Degree Global Tropospheric Analyses and Forecast data product (NCEP, 2015, date of last access: 23.2.2022), and the OSI-SAF L3 Global Sea Ice Concentration product (OSI-SAF, 2017, date of last access: 17.3.2022) is used for the assessment of the residence time of air parcels of sea ice as detailed in Section 2.6.

At NM, the bromide and sodium content of aerosol particles has been analysed by ion chromatography from continuous
filter samples using a ventilated electropolished stainless steel inlet stack. Starting in March 1983 through 2011, a high volume sampling device with a temporal integration of 7 to 14 days was used. Here we only use data from 2012 on. Since that time, a low volume sampling device has been used exclusively. From the in-line air stream, aerosol was continuously sampled using a 2-stage filter system, consisting of a teflon and a nylon filter connected in series. Roughly $60\,\mathrm{m}^3$ of ambient air was probed over a typical collection period of 24 hours. It is important to note that, apart from particulate bromide, also an unknown
amount of gaseous reactive bromine can be absorbed by the filters. Thus, gaseous and particulate bromine compounds cannot be distinguished unequivocally by this operational analysis of the filter samples, which would require a dedicated and more detailed procedure (e.g., Legrand et al., 2016).

## 2.4 DOAS analysis

Spectral analysis and retrieval of BrO and aerosol extinction vertical profiles were already described elsewhere (Frieß et al.,
2011) and are therefore only briefly summarised here. Differential slant column densities (dSCDs), i.e. integrated trace gas concentrations along the light path, of BrO and the oxygen collision complex $O_4$, which serves as a proxy for the light path through the atmosphere and thus for the abundance of aerosols, have been determined from scattered light spectra using the DOASIS software developed at IUP Heidelberg (Kraus, 2006). Absorption cross sections of trace gases with significant optical depth in the respective wavelength region are fitted to the logarithmic ratio of a spectrum measured off-axis (elevation angle
$\alpha < 90°$) to a zenith-sky spectrum ($\alpha = 90°$). According to the Beer-Lambert law, the fit coefficients yield the differential slant column densities $dS_i = \int \rho_i(s)ds - S_{\mathrm{ref}}$, where $\rho_i(s)$ represents the concentration of trace gas $i$ at location $s$, and the integral is performed over all possible light paths from the top of the atmosphere to the instrument. $S_{\mathrm{ref}}$ represents the slant column density contained in the reference spectrum. The literature absorption cross sections included in the BrO and $O_4$ spectral analysis are listed in Table 2. In addition, polynomials of second and third order included in the $O_4$ and BrO fit, respectively,
remove spectral broad-band components caused by Rayleigh and Mie scattering in the atmosphere. A Ring spectrum (Grainger





**Table 2.** Absorption cross sections used for the retrieval of BrO and $O_4$.

| Trace gas | T[K] | Reference |
|---|---|---|
| BrO | 228 | Wilmouth et al. (1999) |
| $NO_2$ | 220, 298 | Vandaele et al. (1998) |
| $O_3$ | 223, 293 | Bogumil et al. (2000) |
| OClO | 233 | Kromminga et al. (2003) |
| $O_4$ | 298 | Hermans et al. (2002) |
| HCHO | 223 | Meller and Moortgat (2000) |

and Ring, 1962; Chance and Spurr, 1997) accounts for the filling-in of Fraunhofer lines due to rotational Raman scattering. Furthermore, the fit allows for a shift and linear squeeze of the measurement spectrum in order to account for changes in the wavelength alignment of the instruments, and a non-linear constant intensity offset accounts for possible instrumental stray light.

The BrO fit window extends from $332.5\,\mathrm{nm}$ to $359.5\,\mathrm{nm}$, encompassing five BrO absorption bands. The $O_4$ dSCDs at the $360\,\mathrm{nm}$ absorption band are retrieved in a fit window ranging from $350\,\mathrm{nm}$ to $370\,\mathrm{nm}$. For each elevation sequence, the mean between the two zenith sky spectra prior and after the sequence, weighted by the time difference to each spectrum of the sequence, serve as Fraunhofer reference. This approach removes the stratospheric signal which is not of interest here.

Histograms of the BrO fit error for both sites are shown in the supplemental Figure S1. The median fit errors for NM and AH
amount to $7.7 \cdot 10^{12}\,\mathrm{molec\,cm^{-2}}$ and $6.0 \cdot 10^{12}\,\mathrm{molec\,cm^{-2}}$, respectively. This amounts to a relative error for measurements at low elevation angles of about 2% in case of a BrO event with a vertical column density (VCD) of $5 \cdot 10^{13}\,\mathrm{molec\,cm^{-2}}$ (corresponding to about 40 ppt of BrO in a $500\,\mathrm{m}$ layer). The relative error of the $O_4$ dSCD is typically in the range of 2%, and represents only a small contribution to the total error of the retrieved aerosol extinction profiles, which is dominated by the smoothing error (Frieß et al., 2006).

**2.5    Retrieval of aerosol and BrO vertical profiles**

Vertical profiles of aerosol extinction $k$ and BrO number concentration $\rho_{\mathrm{BrO}}$ (which are later converted to BrO VMR profiles) are retrieved from MAX-DOAS measurements of dSCDs of $O_4$ and BrO, respectively, using the HEIPRO profile retrieval algorithm already described elsewhere (Frieß et al., 2006, 2011; Frieß et al., 2016, 2019). Based on the well-known optimal estimation method (OEM) (Rodgers, 2000), the algorithm determines the most probable atmospheric state (here: aerosol ex-
tinction and BrO concentration vertical profiles) by comparison of measured MAX-DOAS dSCDs with dSCDs modelled by the SCIATRAN radiative transfer model (Rozanov et al., 2014). The limited information content of the measurements requires additional constraints, which are provided by an a priori vertical profile and the according a priori covariance matrix. The a priori covariance matrix $\mathbf{S}_a$ contains the square of the a priori error in each layer as diagonal elements. Non-diagonal elements



**Table 3.** Main settings of aerosol and BrO vertical profile retrievals.

| | NM | | AH | |
|---|---|---|---|---|
| | Aerosols | BrO | Aerosols | BrO |
| a priori surface value | $0.1\,\mathrm{km}^{-1}$ | $5 \cdot 10^8\,\mathrm{molec\ cm}^{-3}$ | $0.1\,\mathrm{km}^{-1}$ | $5 \cdot 10^8\,\mathrm{molec\ cm}^{-3}$ |
| a priori scale height | $0.7\,\mathrm{km}$ | $0.6\,\mathrm{km}$ | $0.3\,\mathrm{km}$ | $0.6\,\mathrm{km}$ |
| a priori error[1] | 100% | 150% | 150% | 150% |
| a priori correlation length | $0.2\,\mathrm{km}$ | $0.1\,\mathrm{km}$ | $0.5\,\mathrm{km}$ | $0.1\,\mathrm{km}$ |
| state vector | $\ln(k)$ | $\ln(\rho_{\mathrm{BrO}})$ | $\ln(k)$ | $\ln(\rho_{\mathrm{BrO}})$ |
| vertical grid[2] | 25, 94, 187, 316, 492, 733, 1064, 1517, 2137, 2987 | | 20 layers, $100\,\mathrm{m}$ height each | |
| time range | 14.01.2003 - 31.12.2021 | | 24.10.2012 - 31.12.2021 | |
| number of profiles | 71891 | | 40719 | |

(1) Error relative to value of a priori profile; (2) Centre altitude of layers in m.

introduce a correlation between different layers and are set to $S_{a,ij} = \sqrt{S_{a,ii}\,S_{a,jj}} \cdot e^{(h_i - h_j)^2/\sigma_a^2}$, with $h_i$ being the centre
height of layer $i$, and with the correlation length $\sigma_a$ determining the degree of vertical smoothing.

The main settings of the aerosol and BrO gas retrievals at both sites are listen in Table 3. A priori profiles exponentially decreasing with altitude are chosen. The state vectors are represented as the logarithm of the aerosol extinction and BrO number concentration profiles, respectively. Through this, negative extinction and concentration values are excluded and the retrieved state can cover a larger range of possible values.

Although a complete elevation scan takes less than 15 minutes, vertical profiles are retrieved on an hourly basis in order to keep the computational effort for the processing of several decades of data on an acceptable level. For this purpose, all dSCDs measured during one hour serve as input for a single retrieval. This means that between 3 and 5 elevation scans contribute to the measurement vector, and the resulting profiles represent an hourly average with variations of the aerosol and BrO amount at shorter time scales not being resolved. Using all measurements during one hour for a single retrieval reduces the noise error
of the retrieved profile, but this has only a minor effect on the total error, which is dominated by the smoothing error (Frieß et al., 2006).

An important diagnostic variable of the OEM retrieval is the averaging kernel (AVK) matrix $A$. It is defined as the sensitivity of the retrieved to the true profile and provides a measure for the vertical resolution of the retrieval. The total information content of the retrieval is given by the degrees of freedom for signal (DOFS) $d_s = \mathrm{Tr}(A)$, which represent the maximum
number of independent quantities that can be inferred from the measurements.

The sensitivity of MAX-DOAS measurements in the near UV quickly decreases with altitude and is usually limited to the lowermost $1.5\,\mathrm{km}$ of the atmosphere above the instrument (Frieß et al., 2011). In order to distribute the information content more evenly over the individual layers, a vertical grid with the layer thickness increasing with altitude was applied for the Neumayer retrieval (see Table 3). Starting with $50\,\mathrm{m}$ at the ground, the layer thickness subsequently increases with height up





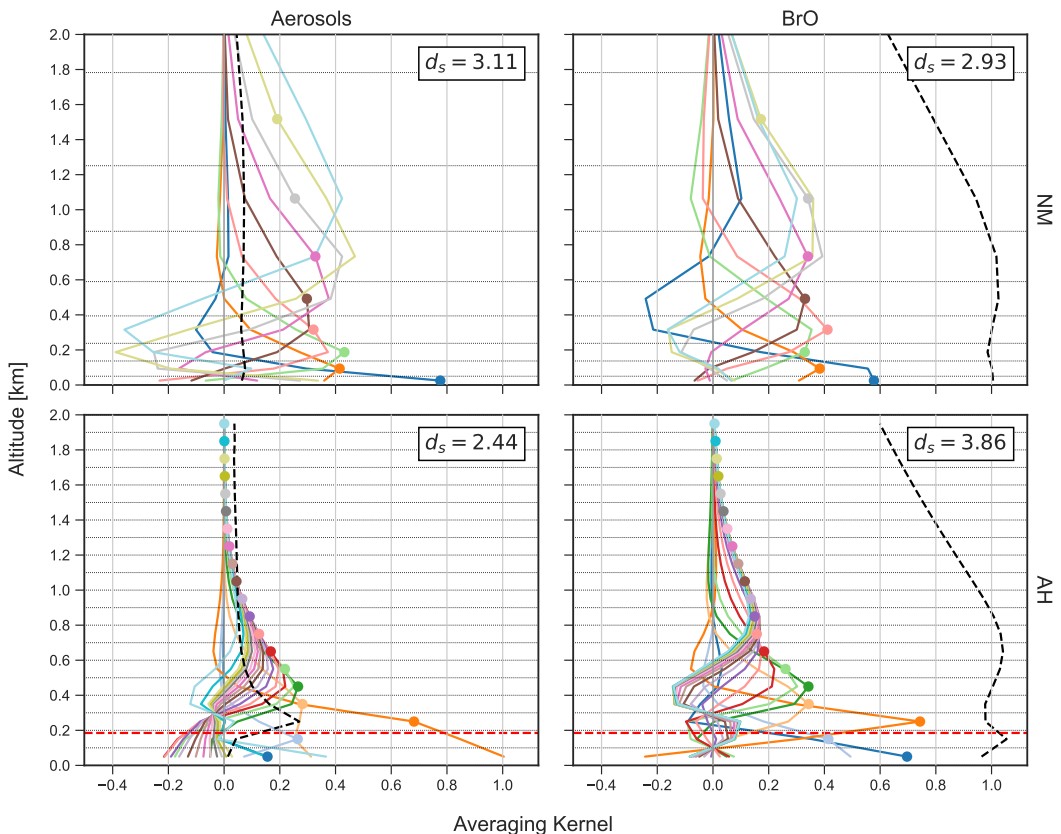

**Figure 2.** Examples of clear-sky AVKs for aerosols (left) and BrO (right). Top panels show AVKs for NM, bottom panels AVKs for AH. Each colour shows the AVK for an individual height layer, with its nominal height indicated by a coloured dot. The dashed lines show the averaging kernels of the total column in units of km. Layer boundaries are shown as dotted horizontal lines. The red dashed lines in the bottom panels indicate the height of the AH instrument over sea level.

to a thickness of 1 km for the layer centred at 2.5 km, as indicated by the dotted horizontal lines in the upper panel of Figure 2. Unfortunately, the application of a non-regular grid led to an unstable behaviour for the retrieval of vertical profiles at AH. This is most probably caused by the fact that the instrument is located at an elevated altitude, which is represented in the radiative transfer model by an instrument being located at a height of 185 m above the ground (dotted red line in Figure 2). Therefore, a regular grid as in traditional MAX-DOAS OEM retrievals was applied for the AH retrieval. The retrieval grid extends from the ground up to an altitude of 2 km and consists of 20 layers with a constant thickness of 100 m each (see Table 3).

The clear-sky AVKs shown in Figure 2 illustrate that the information is distributed more evenly over the different layers for the irregular grid (NM, top panels) than for the regular grid (AH, bottom panels). The AVKs for NM at the surface have peak values of 0.8 and 0.6 for aerosols and BrO, respectively, and remain at a nearly constant value of 0.4 for the layers above. BrO AVKs peak near their nominal altitude, indicating that the information comes from the nominal height level. In contrast, NM





AVKs for aerosols at altitudes above 600 m peak below their nominal height, indicating that parts of the extinction retrieved at higher levels actually originate from lower altitudes.

The sensitivity of AH retrievals performed on a regular grid is restricted to lower altitudes, with little information on the atmosphere above 1 km and 1.5 km for aerosols and BrO, respectively. Interestingly, the sensitivity near the instrument altitude and below is quite different for aerosols and trace gases. Using the additional downward viewing direction, the BrO retrieval

has good sensitivity for trace gases at low altitudes right above the sea ice, with an AVK peak value of the surface layer of 0.7. Furthermore, there is good sensitivity for the (200-300) m layer directly above the instrument, where the according AVK shows a marked peak. The sensitivity of the AVK of the (100-200) m layer, in which the instrument resides, is comparably low with a peak value of 0.4. In contrast to BrO, the sensitivity of the aerosol retrieval to the lowermost two layers is very low (around 0.2). The fact that the (200-300) m layer does not peak at its nominal altitude but at the surface indicates that the

aerosol extinction near the surface is falsely attributed to this layer.

The averaging kernels of the total column, quantifying the sensitivity of the total column to variations of the extinction or concentration in each layer, are shown as dashed lines in Figure 2. The sensitivity of the BrO VCD is very similar for AH and NM, whereas there is a peak in sensitivity for aerosols in the layer right above the instrument at AH that is not present at NM. The AH aerosol total column AVK is close to zero below the instrument indicating a lack of sensitivity to aerosols in this

altitude range.

Histograms of the DOFS including all data are shown in the supplemental Figure S2. The histograms for NM show two peaks because an additional 1° viewing direction introduced in February 2016 increased the information content by about half a unit both for aerosols and BrO. After including the 1° elevation angle, the information content of aerosol extinction profiles is slightly better for NM ($\bar{d}_s = 2.89$) than for AH ($\bar{d}_s = 2.63$). However, the peak of the distribution of DOFS of BrO profiles

is more than half a unit higher for AH ($\bar{d}_s = 3.51$) than for NM ($\bar{d}_s = 2.93$). This illustrates that observations from elevated sites (mountains, towers or tall buildings) and the inclusion of downward viewing directions greatly enhances the sensitivity of MAX-DOAS measurements for atmospheric trace gases compared to measurements from the ground.

### 2.6 Trajectory modelling and source-receptor analysis

One of the main objectives of this study is the investigation of potential source regions for BrO and the influence of aerosols

and meteorological parameters. This is done using a source-receptor analysis that is based on the approach presented by Frieß et al. (2004) and later applied for the interpretation of long-term measurements in the Arctic by Bognar et al. (2020). 72 h back trajectories ending at each of the two measurement sites are calculated using the HYSPLIT model with GDAS meteorological data. Back trajectories are calculated for each hour, with end point altitudes corresponding to the centre altitudes of the MAX-DOAS retrieval grids. An example for back trajectories ending at NM in the morning of 10. September 2016 is shown in Figure

3. Also shown in this plot is the sea ice concentration $\rho_{\text{ice}} \in [0, 1]$ from the OSI-SAF data product. For each of the trajectories, contact times of the respective air parcel with sea ice ($\tau_{\text{ice}}$), open ocean ($\tau_{\text{water}}$), and land ($\tau_{\text{land}}$, including the Antarctic ice sheet and the shelf ice regions), as well as the residence time in the free troposphere ($\tau_{\text{free}}$) have been determined. The effective residence times over sea ice and open water are calculated by weighting the residence time with $\rho_{\text{ice}}$ and $(1 - \rho_{\text{ice}})$,



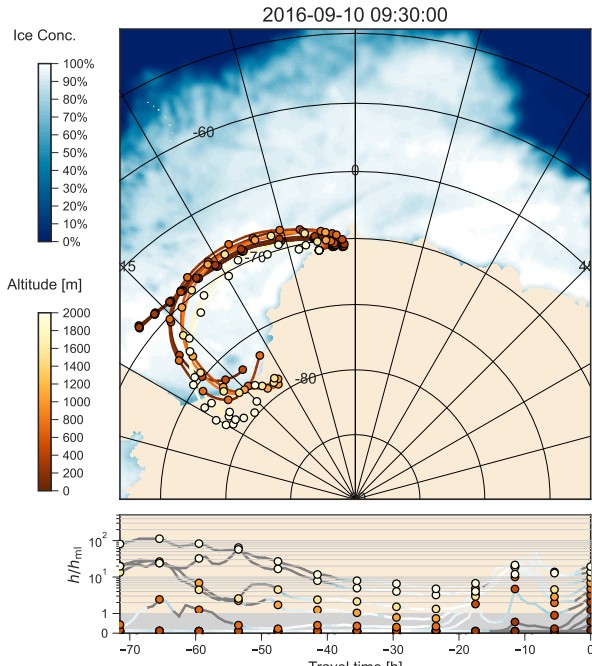

**Figure 3.** Example for back trajectories ending at Neumayer Station on 10.9.2016 at 9:30 am. The trajectories end at the centre of the profile retrieval layers. In the top panel, the altitude of the air parcels above ground level (AGL) are given by the line colour (red to white colour scale), and each circle indicates 6 hours of travelling time, with the circle colour indicating the end point altitude. Blue to white colours indicate the OSI-SAF sea ice concentration (Copyright ©2018 EUMETSAT) as denoted in the respective colour bar. The bottom panel shows the time-height cross section, with the height being normalised to the mixing layer height. The line colour indicates the sea ice concentration below the air parcel (blue to white), with locations over the continent shown in grey. Note the change in y-axis scale from linear between 0 and 1 (grey area, corresponding to the vertical extent of the mixing layer) to logarithmic above.

respectively. An air parcel is considered to be in contact with the surface if its altitude is below the mixing layer height (MLH) $h_{\mathrm{ml}}$. The MLH is a quantity estimated by the HYSPLIT model and serves as an estimate for the boundary layer height. This way, hourly vertical profiles of contact time of the air parcels above the measurements sites with these different surface types are constructed and can be related to the corresponding vertical profiles of BrO and aerosols from MAX-DOAS.

An example for the results of the trajectory analysis during a period of enhanced BrO at NM in September 2016 is shown in Figure 4. On 10.9.2016, about 20 ppt of BrO are present at the surface, and the BrO layer extends over an altitude of $\approx 1\,\mathrm{km}$ above the surface. The vertical profile of contact time with sea ice shows a large degree of similarity to the shape of the BrO profile from MAX-DOAS, while contact times with land are very small ($< 8\,\mathrm{h}$) and there was very little contact with water surfaces. More detailed case studies on the relationship between BrO, aerosols and airmass history will be presented in Section





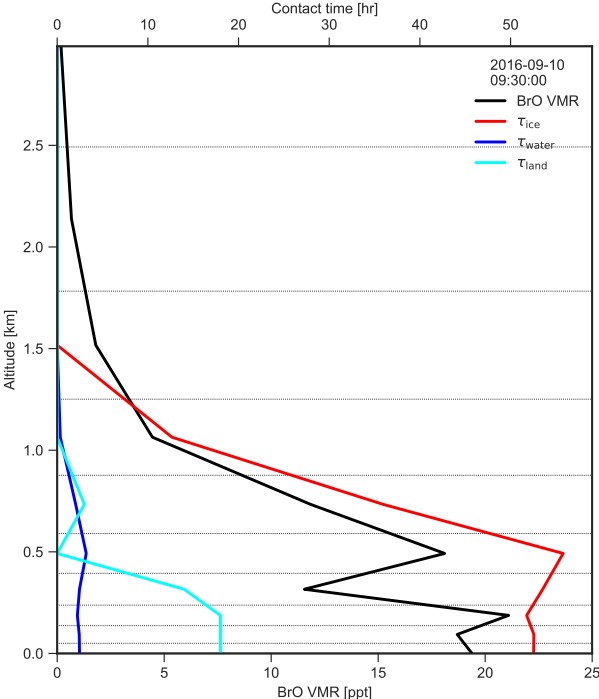

**Figure 4.** Example for a vertical profile of BrO at NM together with the contact times of the air parcels over ice, open water and land on 10 September 2016, 09:30 UTC. Dotted horizontal lines indicate the retrieval layer boundaries.

3.1., and a statistical analysis of the relationship between different meteorological and geophysical parameters observed in situ and along the trajectories will be discussed in Section 3.3.

To identify potential source regions for reactive bromine, a source-receptor analysis was performed for BrO, aerosols as follows: First, the BrO VMR and aerosol extinction observed at the observation site was assigned to the respective air parcel of the back trajectory simulations. Then the median BrO VMR and aerosol extinction of all air parcels passing over each $1° \times 1°$ latitude/longitude bin in an altitude below the MLH was calculated. By using the large number of observations from our long-term measurements, this yields maps of the source-receptor relation for BrO and aerosols in the Antarctic region. In addition,

the mean wind speed and direction in each grid cell for air parcels located below the MLH are calculated from the trajectory data. The resulting source-receptor maps will be discussed in Section 3.4.

### 2.7    Regression analysis

The large number of MAX-DOAS measurements with more than $100\,000$ pairs of BrO and aerosol vertical profiles was used to infer the relationship between BrO and other parameters observed at both stations and modelled along the trajectories. The

Pearson correlation coefficient (PCC) $R$ for each pair of parameters observed at the measurement sites serves as a basis to investigate the relationship between the available meteorological and chemical quantities. To better account for possible non-





**Table 4.** Surface parameters included in the regression analysis.

| Parameter | Symbol | Source |
| --- | --- | --- |
| BrO VMR at surface | BrO | MAX-DOAS profile |
| Aerosol extinction at surface | Aer | MAX-DOAS profile |
| Ozone VMR | $O_3$ | Ozone monitor |
| Particulate bromide | $Br^-$ | Aerosol filter samples |
| Bromide to sodium ratio | Br/Na | Aerosol filter samples |
| Wind speed | $v$ | Meteorological observations |
| Atmospheric pressure | $p$ | Meteorological observations |
| Relative humidity | RH | Meteorological observations |
| Temperature | $T$ | Meteorological observations |
| Temperature gradient (10 m - 2 m) | $\Delta T$ | Meteorological observations |

**Table 5.** Columnar parameters included in the regression analysis.

| Parameter | Symbol | Source |
| --- | --- | --- |
| BrO VMR profile | BrO | MAX-DOAS profile |
| Aerosol extinction profile | Aer | MAX-DOAS profile |
| Ozone profile | B-$O_3$ | Ozone balloon sonde |
| Contact time with sea ice | $\tau_{ice}$ | Hysplit and OSI-SAF |
| Wind speed over sea ice | $v_{ice}$ | Hysplit and OSI-SAF |
| Contact time with sea water | $\tau_{water}$ | Hysplit and OSI-SAF |
| Contact time with land | $\tau_{land}$ | Hysplit and OSI-SAF |
| Residence time in free troposphere | $\tau_{free}$ | Hysplit and OSI-SAF |
| Integrated radiation along trajectory | RAD | Hysplit |
| Mean temperature along trajectory | $T_{traj}$ | Hysplit |
| Mean mixing layer height over ice | $h_{ml}$ | Hysplit and OSI-SAF |

linear effects, the PCC of BrO VMR as well as aerosol was determined using the logarithm of both parameters. The regression analysis was performed separately for the months August/September/October (ASO), November/December/January (NDJ), as well as February/March/April (FMA).

The parameters observed at the surface level that are included in the regression analysis are listed in Table 4. Apart from BrO and aerosol extinction retrieved at instrument altitude, these include the most important meteorological parameters (wind, temperature, pressure, relative humidity) as well chemical parameters and as in situ ozone. The columnar parameters listed in





Table 5 include BrO and aerosol extinction profiles from MAX-DOAS, ozone VMR from balloon soundings averaged over the
MAX-DOAS retrieval layers, contact times with different surface types inferred from the back trajectory analysis, as well as
integrated radiation, mean temperature and mean MLH over ice along the trajectories which are calculated by the HYSPLIT
model on the basis of GDAS meteorological fields. In order to exclude MAX-DOAS data with insufficient information content,
the statistical analysis includes only BrO and aerosol profiles with a DOFS of $d_s > 1.75$. This means that measurements during
extremely low visibility, in particular blowing snow, are excluded. This way, about 24% and 19% of the profiles at NM and
AH, respectively, are discarded. The results of the regression analysis will be presented in Section 3.3.

## 3    Results and Discussion

### 3.1    Relationship between BrO, Aerosols and Air Mass History: Case Studies

Examples for the relationship of BrO, aerosols, air mass history, meteorology and ozone abundance during BrO enhancement
episodes at NM and AH are presented in Figures 5 and 6, respectively.

The BrO enhancement episode at NM during August/September 2006 shown in Figure 5 lasts for 12 days. Between 28.
August and 8. September, air masses above NM were in contact with sea ice prior to their arrival at the measurement site
for most of the time. At the same time, enhanced BrO was detected by the MAX-DOAS instrument. On 28. August, the
lowermost $3\,\mathrm{km}$ of the atmosphere were in contact with sea ice for a more than $30\,\mathrm{h}$, while the MAX-DOAS instrument
simultaneously detects enhanced BrO in the same altitude range. The surface layers are, however, less affected by sea ice
contact, and contain less BrO than the layers above. On 29. August, an uplifted layer of BrO with a peak altitude around $1\,\mathrm{km}$
builds up in coincidence with air masses with significant sea ice contact at slightly higher altitudes than on the previous day.
The difference between the shape of the sea ice contact time profile and the BrO profile can be attributed to the lower sensitivity
of MAX-DOAS to higher altitudes, which is even more limited by the presence of enhanced aerosol extinction near the surface,
probably due to blowing snow at wind speeds close to $20\,\mathrm{m\,s^{-1}}$. Ozone is slightly depleted during this period, dropping from
a background value of about 32 ppb to less than 25 ppb.

The period between 2. and 5. September is characterised by low aerosol extinction at slightly lower wind speeds than before.
During this time, a shallow layer of enhanced BrO with a vertical extent of only 500 - $800\,\mathrm{m}$ is present. The trajectory analysis
reveals that only these air masses close to the surface were previously in contact with sea ice. At the same time, ozone is
significantly depleted with minimum values below 10 ppb in the night from 3. to 4. September.

Between 6. and 8. September, wind speeds of up to $30\,\mathrm{m\,s^{-1}}$ lead to blowing snow with reduced visibility and extinction
coefficients of more than $10\,\mathrm{km^{-1}}$. On 7. and 8. September, air masses with sea ice contact times of more than 30 hours are
present at altitudes above $1.5\,\mathrm{km}$. At the same time, uplifted BrO layers with peak altitudes around 1 km and BrO VMRs above
10 ppt are retrieved, despite the fact that the averaging kernels indicate a very low sensitivity to these altitudes owing to the
low visibility that results in large fluctuations of the BrO vertical profiles. Thus it is quite possible that the true BrO layer was
at higher altitudes than the retrieval indicates. At the same time, there is only little ozone depletion with the $O_3$ VMR being
340 reduced by less than 5 ppt, providing further evidence that the detected BrO resides in an uplifted layer.





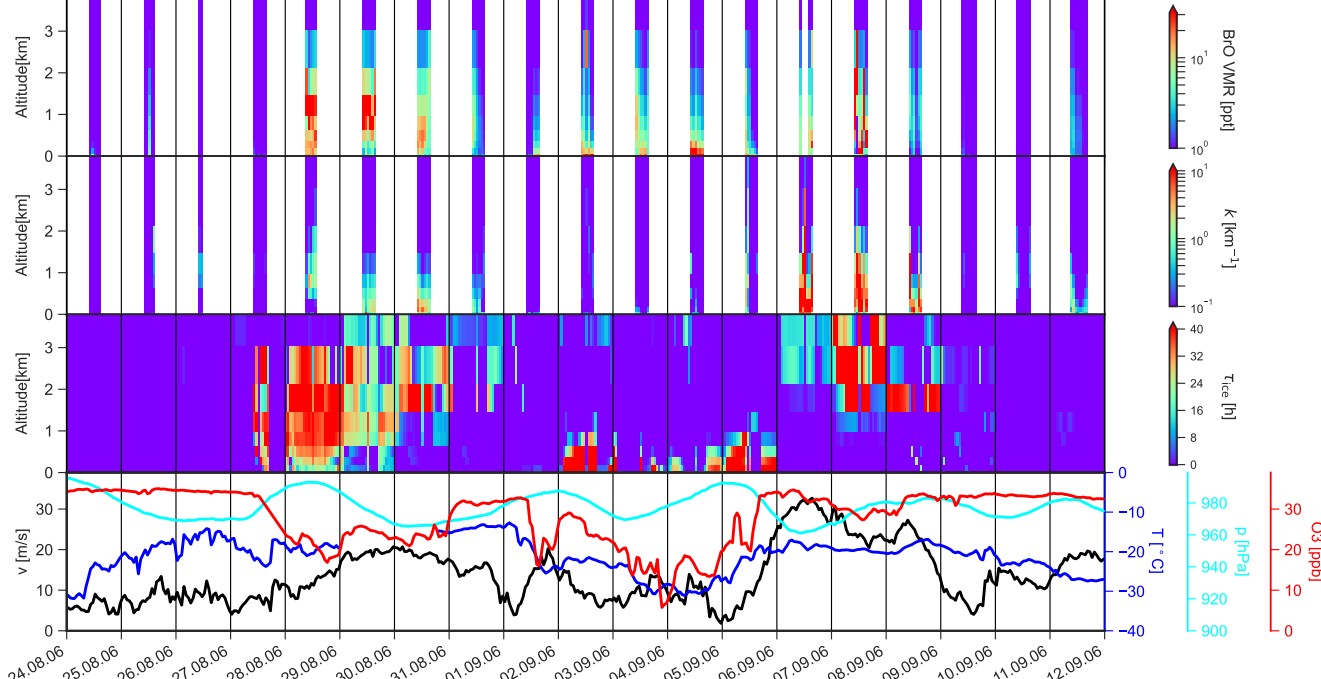

**Figure 5.** Example for a BrO enhancement episode at NM during August/September 2006, showing (from top to bottom) (1) BrO VMR profiles, (2) aerosol extinction profiles, (3) profiles of the duration of sea ice contact as well as (4) wind speed (black), surface temperature (blue), pressure (cyan), and surface ozone VMR (red). Note the logarithmic colour scale of the BrO and aerosol extinction profile. Date/Time is given in UTC.

The wind speed decreases and the visibility increases in the afternoon of 8. September. Afterwards, the air masses above NM were not in contact with sea ice any more and no BrO is detected above the detection limit, while the ozone VMR remains at values of 33 ppb, representing typical background levels for this time of the year.

The BrO enhancement episode at AH during September 2015 shown in Figure 6 illustrates that BrO is usually distributed
345 more diffusely and over a larger altitude range than at NM. This is a result of the more complex topography around AH, with the slopes of Ross Island leading to advection and vertical mixing of air masses. The period shown here is characterised by low temperatures, ranging between -20°C and -40°C, and wind speeds below $20\,\mathrm{m\,s^{-1}}$. The BrO enhancement episode is initiated on 9. September by air masses in the lowermost $1\,\mathrm{km}$, i.e. mostly below the MAX-DOAS instrument, which were in contact with sea is for a relatively short duration of about 15 h. At the same time, BrO VMR of 5-10 ppt are detected by MAX-DOAS in
350 the lowermost $400\,\mathrm{m}$ above the sea ice. Longer sea ice contact times in the same altitude range between 11. and 13. September lead to BrO VMRs above 10 ppt, distributed over an altitude range of 1.5-2 km, coincident with periods of enhanced aerosol extinction near the ground from 12. September onwards.



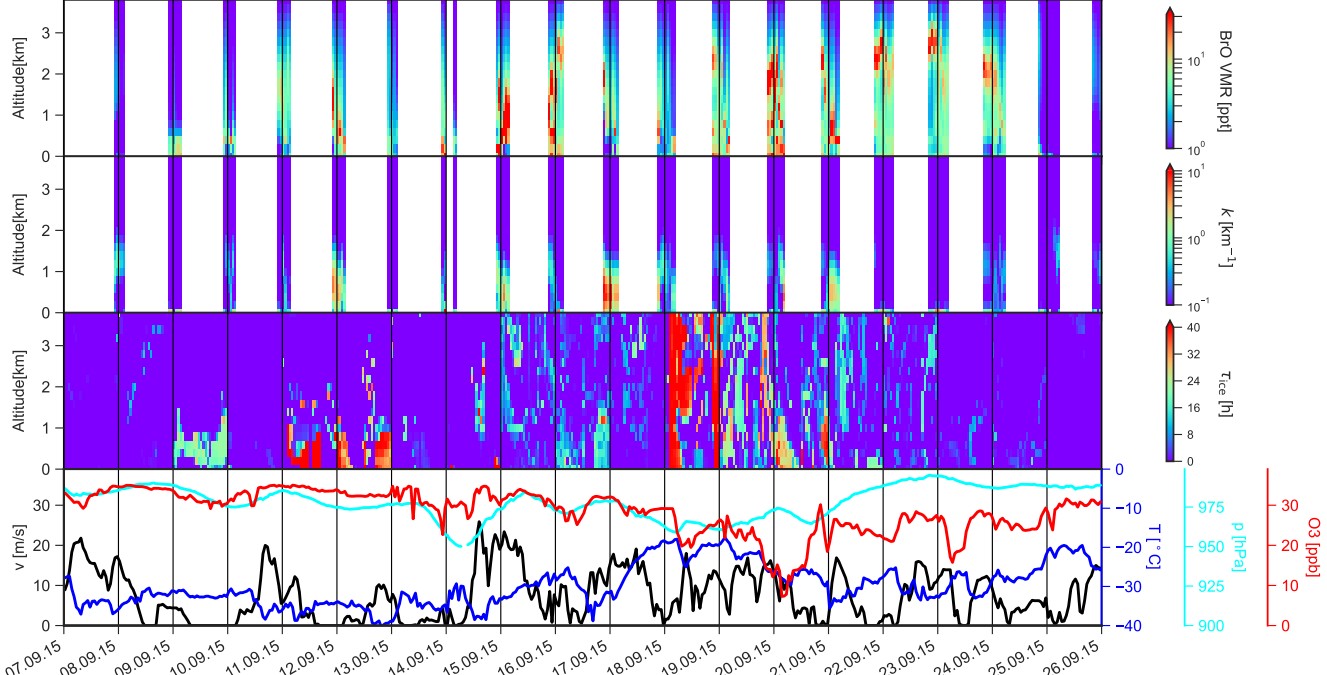

**Figure 6.** Same as Figure 5, but for a BrO enhancement episode at AH during September 2015. Note that meteorological parameters and ozone VMR are measured at the altitude of the MAX-DOAS instrument (184 m ASL).

The wind speed increases after 14. September and air masses with contact to the sea ice are present in the entire lower troposphere. BrO is found to be almost evenly distributed over the lowermost 3 km, despite the fact that the sensitivity of the MAX-DOAS retrieval is low at high altitudes. During this period, BrO VMRs exceed 20 ppt and ozone starts to decline continuously until it reaches a minimum of 7 ppb on 20. September. Afterwards, there is only little contact of the air masses near the ground with sea ice and ozone slowly recovers to reach background values of 30 ppb on 25. September.

An uplifted BrO layer at almost 3 km altitude, with peak values of VMR above 20 ppt, is present between 21. and 24. September, while around 2-5 ppt of BrO are present between the peak altitude and the surface. The trajectory analysis reveals that these air masses enriched in BrO had sea ice contact times of 15 - 20 hours. A continuous decrease in the BrO peak altitude between 23. and 25. September is coincident with a similar decrease in the altitude range of air masses that were previously in contact with sea ice.

## 3.2 Spatio-temporal Variability of BrO

The seasonal variation of BrO VCD and AOD is presented in Figure 7. As expected, highest BrO amounts are found during austral spring, with a maximum right after polar sunrise in August. During spring, BrO VCDs at NM are significantly larger than at AH, and occasionally exceed $1 \cdot 10^{14}$ molec cm$^{-2}$. The $95^{th}$ percentile of the BrO VCD at NM is about three times




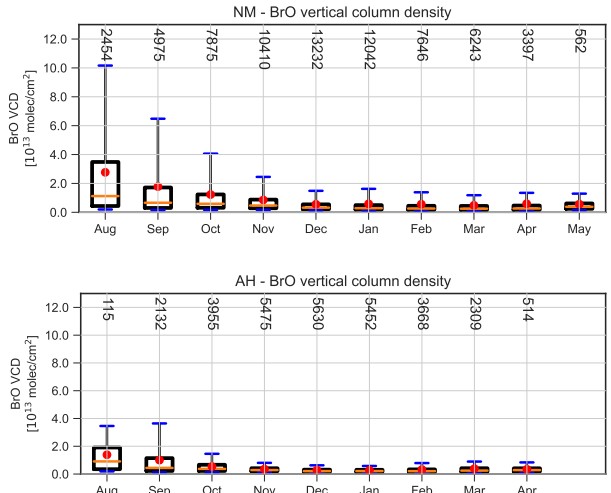

**Figure 7.** Box-whisker-plots of the seasonal variation of the BrO VCD for NM (top) and AH (bottom). Here and in all following box-whisker-plots, boxes indicate the $(25 - 75)^{th}$ percentile and whiskers the $(9 - 95)^{th}$ percentile. Mean and median are shown as red dots and vertical orange lines, respectively. The numbers denote the number of data points in each bin.

larger than at AH. It can be excluded that these differences between both sites are due to the different vertical sensitivity of the profile retrieval because the averaging kernels for the BrO VCD are of similar magnitude (see Figure 2). The larger abundance of BrO at NM compared to AH is instead caused by the differences in meteorological conditions, with the majority of the air masses at AH coming from the interior of the continent while those at NM are mainly of maritime origin. Starting at polar sunset in August, the distribution of the BrO VCDs over the year follows an exponential-like decay at both stations.

In contrast to AH, significant amounts of BrO are also present at NM during summer and autumn (December to May), with more than 5% of the observations exhibiting BrO VCDs of more than $1.5 \cdot 10^{13} \, \mathrm{molec \, cm^{-2}}$. The processes leading to the presence of BrO have been studied in detail by Nasse (2019) on the basis of continuous multi-annual longpath-DOAS measurements at NM. It was found that BrO release during summer usually occurs under very calm and stable conditions with strong temperature inversions near the surface, which enable the accumulation of reactive bromine and lead to BrO VMRs of sometimes more than 100 ppt. These events are usually preceded by stormy periods, during which deposition of blowing snow, ice particles and/or sea spray takes place. This provides evidence for the hypothesis that saline particles from the sea ice and/or from the open ocean transported to the measurement site on the shelf ice lead to the release of reactive bromine from the snow surface in situ. Only very little ozone destruction is observed during these BrO events in summer, since their duration ranges only from several hours to at most three days. This situation is very different to the commonly known BrO explosions during spring, when large-scale transport of reactive bromine released from the sea ice to the measurement site takes place, leaving sufficient time for the depletion of ozone in the observed air masses. In contrast to NM, no significant amounts of BrO are



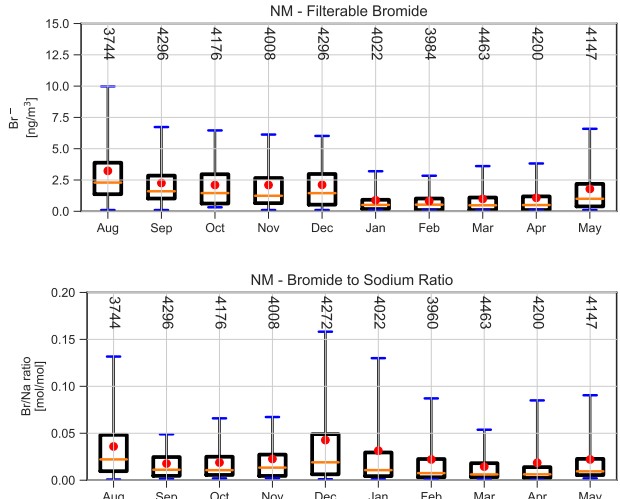

**Figure 8.** Box-whisker-plots of the seasonal variation of the particulate bromide concentration (top) and bromide to sodium ratio (bottom) determined from aerosol filter samples.

detected at AH during summer and autumn. Again, this can be explained by the different meteorological conditions with air
masses coming mainly from the interior of the Antarctic continent.

The bromide concentration derived from filter samples at NM shown in Figure 8 has its maximum between August and December, remaining at a constant level during this period. There is a sudden decline in bromide from December to its minimum value in January, when most sea ice has disappeared. Particulate bromide concentrations increase again as the sea ice reforms during autumn, while the BrO abundance remains low. The molar bromide to sodium ratio in aerosols shown in the
bottom panel of Figure 8 is strongly enhanced compared to mean sea water with a ratio of only $1.8 \cdot 10^{-3}$. This enhancement is in agreement with surface snow measurements on the Greenland ice sheet (Dibb et al., 2010), whereas snow sample measurements in the Arctic coastal region show $Br^-/Na^+$ ratios similar to the ocean water or even depleted in bromide (Simpson et al., 2005). The bromide to sodium ratio is smallest in the months September and October, as well as March and April, and has a maximum in August and December. This seasonality can either be driven by a depletion of particulate bromide by het-
erogeneous release of reactive bromine on aerosols, by fractionation processes during the formation of the saline surfaces from which the particles originate, or by changes in the origin of the particles (sea ice versus open ocean) in the course of the year. In summary, it is not obvious from a seasonal perspective whether there is a direct link between bromine on airborne particles and BrO, which could point towards a heterogeneous release of reactive bromine from aerosols. More substantial conclusions will be drawn from the regression analysis presented in Section 3.3, where the relationship between particulate bromide and
BrO is investigated separately for each season.

The vertical distribution of the BrO VMR as a function of SZA during periods of enhanced BrO, quantified by the $95^{th}$ percentile of the BrO VMR, is shown in Figure 9. During ASO, BrO shows a pronounced diurnal cycle with distinct maxima




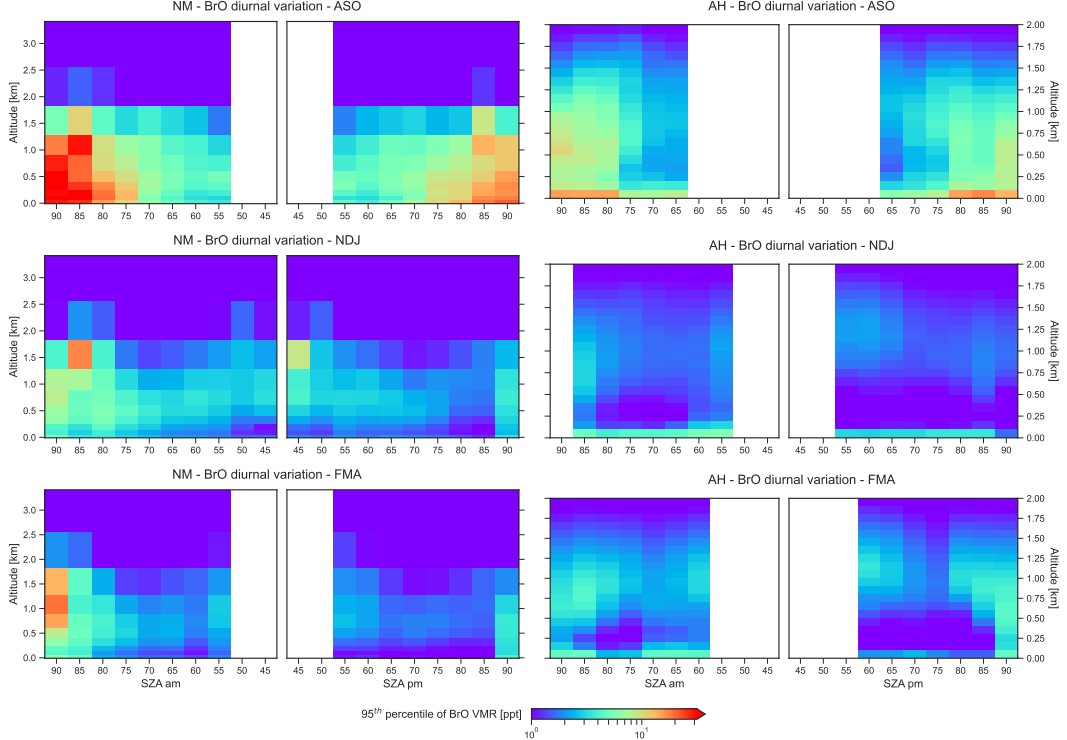

**Figure 9.** The vertical distribution of the BrO VMR in the lowermost 3.5 km during periods of enhanced BrO as a function of SZA for NM (left column) and AH (right column) during ASO (top), NDJ (middle) and FMA (bottom). The colour code indicates the $95^{th}$ percentile of the BrO VMR in each season.

after sunrise and before sunset. The observed BrO minimum at noon is likely to be caused by an interplay of dynamics, with increased vertical mixing around noon, and photochemistry with a shift of the partitioning between Br and BrO towards Br via reaction R1 in the presence of higher solar radiation. Higher BrO amounts are observed in the morning than in the evening at NM, whereas the diurnal variation at AH is more symmetrical. The morning peak in BrO is even more pronounced at NM during NDJ and FMA.

The morning time maximum at NM is probably caused by the nocturnal release of molecular bromine ($Br_2$) from the snowpack by reaction of ozone with bromide (Reactions R5 and R6), which is rapidly converted to reactive bromine via photolysis by visible light during sunrise. The fact that the strongest BrO morning peak is observed during FMA, when very little sea ice is present, indicates a local release of $Br_2$ after deposition of saline particles, such as sea spray, on the snowpack during storms, a process that has already been observed form our LP-DOAS measurements at NM (Nasse, 2019).

The fact that the BrO abundance at AH is much smaller and does not show any diurnal variability during NDJ and FMA indicates that local sources are of less importance. This can be explained by the fact that deposition rate of saline particles (sea





415 spray or blowing snow dispersed from the sea ice surface) is expected to be much lower than at NM since air masses mostly come from the interior of the Antarctic continent.

The vertical distribution, and in particular the presence of uplifted layers of BrO, are of particular interest since elevated BrO concentrations aloft are an indication for a possible transport of reactive bromine into the free troposphere, where already small amounts are expected to have a significant impact on ozone and cloud formation and thus on climate (von Glasow et al.,

420 2004; Roscoe et al., 2014).

As shown in Figure 9, the vertical distribution of BrO significantly changes over the year and is quite different at both sites. During ASO, BrO is equally distributed over the lowermost 1.5 km of the atmosphere at NM. In contrast, at AH a BrO maximum is present at the surface, and BrO is distributed over the lowermost 2 km with a lower abundance than at NM, probably because the local topography leads to enhanced vertical mixing.

425 During NDJ and FMA, BrO is present in a shallow layer at altitudes below 100 m at AH. In addition, a second BrO layer is present aloft at altitudes between 750 m and 1500 m. The trajectory analysis indcates that at least parts of these uplifted BrO layers can be attributed to long-range transport of BrO-rich air parcels, with the air occasionally coming from the WS and being transported over the Antarctic ice sheet to the RS. These transcontinental transport processes will be further discussed in Section 3.5. During NDJ and FMA, BrO is also present aloft at NM, where a layer low in BrO with a maximum thickness

430 of about 300 m develops in the course of the day. This indicates that the snow surface on the ice shelf around NM acts as a sink for reactive bromine during the day due to insufficient amounts of bromide in the surrounding snowpack and/or dilution by vertical mixing, which means that Reaction R4 does not lead to an amplification of gaseous reactive bromine compounds. However, during the night the surface acts as a source of gaseous bromine when $Br_2$ accumulates in the boundary layer owing to release by heterogeneous reaction of bromide with ozone according to Reactions R5 and R6.

435 We further investigate the vertical distribution of BrO based on two key quantities, namely the BrO layer width $w$ and the relative layer height $\tilde{h}$, which are defined as follows: First, a Gaussian distribution is fitted to the BrO profile, yielding its centre height $h$ and width $\sigma$. If the centre of the Gaussian distribution is above the surface ($h \geq 0$), then the width is defined either as $w = \sigma$ if $h \geq \sigma$, or as $w = (h + \sigma)/2$ if $h < \sigma$. The relative layer height is then defined as $\tilde{h} = h/\sigma$, and a profile can be considered as being uplifted if $\tilde{h} > 1$. If the centre of the fitted Gaussian is below the surface ($h < 0$), then the layer width is

440 defined as the e-folding height of an exponential fit to the profile, and the relative layer height is set to zero. Only observations with a BrO VCD $> 1 \cdot 10^{13}$ molec cm$^{-2}$ are considered for the calculation of $\tilde{h}$ and $w$.

The distribution of the layer width $w$ shown in Figure 10 ranges between 0 and 1 km for NM, but reaches higher values of up to 1.5 km for AH, again indicating that there is stronger vertical mixing at AH. Very shallow layers with a vertical extent of less than 100 m are most frequently observed during ASO, while the layer width is larger during NJD and FMA, when shallow

445 layers with widths below 200 m are much less frequent, probably owing to enhanced vertical mixing in summer.

The distribution of the relative layer height $\tilde{h}$ shown in Figure 10 confirms that the majority of the profiles resides at the surface during ASO, while uplifted profiles with a relative height $\tilde{h} > 1$ become more frequent during NDJ, and in particular during FMA. This represents a strong indication for transport of BrO into the free troposphere, where already small amounts of



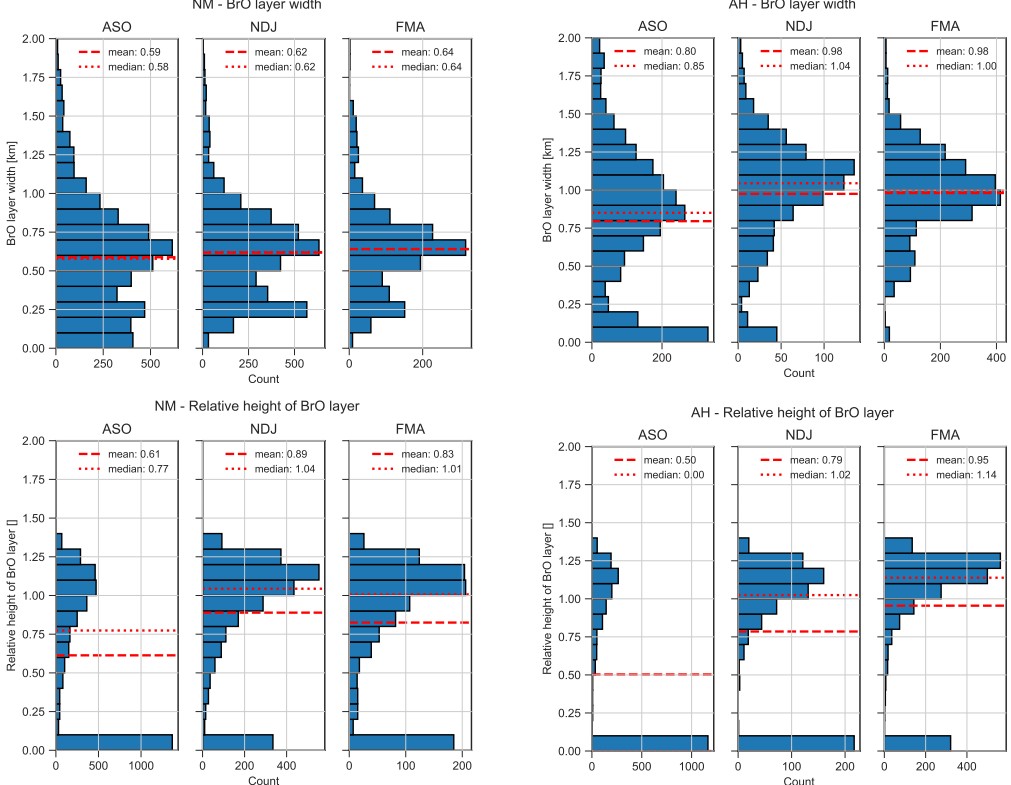

**Figure 10.** Histograms of BrO layer width $w$ (top) and relative layer height $\tilde{h}$ (bottom) for NM (left) and AH (right) during ASO, NDJ and FMA (for a definition of $w$ and $\tilde{h}$ see text). Dashed and dotted horizontal lines indicate mean and median, respectively.

reactive bromine are expected to have a significant impact on the ozone budget as well as on the oxidation of dimethyl sulfide

(DMS) and thus on cloud formation (von Glasow et al., 2002a).

### 3.3 Regression Analysis

In this Section, the relation between different chemical, physical and meteorological quantities available at both measurement sites is discussed on the basis of a regression analysis as described in Section 2.7. Apart from the PCC for each pair of parameters, which quantifies the degree of correlation ($R > 0$) or anti-correlation ($R < 0$), we also inspected the corresponding

p-values (shown in supplemental Figure S9), which represent the probability for the hypothesis that two parameters are not correlated. Apart from a few exceptions, p-values are very close to zero, indicating that the probability of a correlation between most pairs of variables is high.

The PCCs for pairs of columnar and surface observables during each season are shown as matrices in Figure 11. Correlation plots for each pair of variables can be found in the supplement (Figures S3 - S8), where also slope and intercept of a linear

fit based on orthogonal distance regression (Boggs et al., 1987) are presented. At both sites, the most persistent correlation





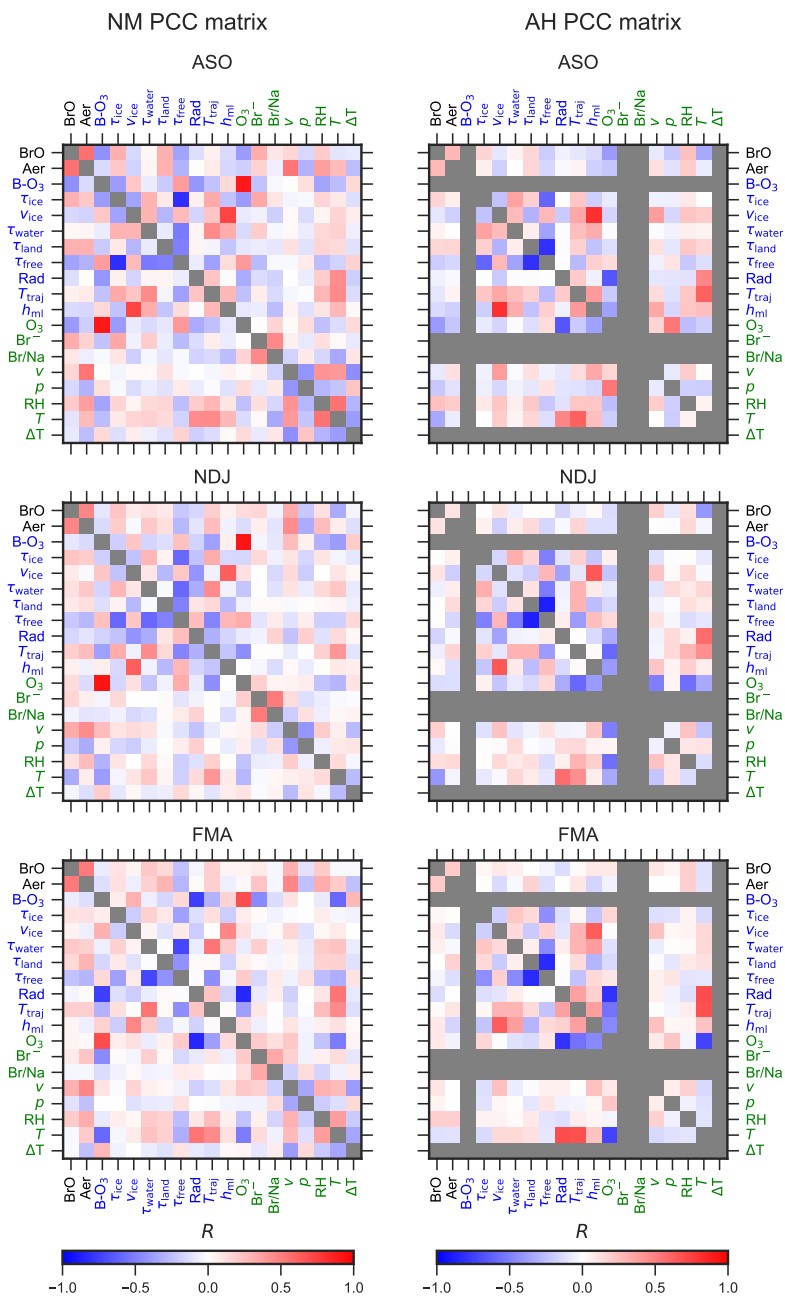

**Figure 11.** Matrix of Pearson's Correlation Coefficients (PCC) between pairs of observed/modelled variables at NM (left) and AH (right) during each season. For a description of columnar and surface parameters see Tables 5 and 4, respectively. Positively correlated parameters are shown in red, anti-correlated parameters in blue. Gray elements indicate unavailable data. Columnar variables related to the trajectory simulations and the ozone soundings are labelled in blue, surface variables observed on site in green.





exists between BrO and aerosols ($0.47 < R < 0.56$ at NM and $0.11 < R < 0.28$ at AH). This provides further evidence for the hypothesis that the release from saline particles, present either as blowing snow or sea salt aerosols, represent one of the most important sources for reactive bromine in polar regions (Frieß et al., 2011; Peterson et al., 2017; Choi et al., 2018; Hara et al., 2018).

At NM, BrO and aerosol extinction are both positively correlated to the duration of sea ice contact during all seasons, with a maximum during ASO ($R = 0.3$ for the correlation between BrO and $\tau_{\text{ice}}$). In contrast, a noticeable positive correlation for BrO with $\tau_{\text{ice}}$ at AH is only found during ASO ($R = 0.17$) when sea ice extent is largest. These findings confirm that the sea ice is a main source for reactive bromine.

Bromide from aerosols filter samples at NM is positively correlated to BrO during all seasons. However, the PCC between

bromide and BrO decreases from 0.32 during ASO over 0.16 during NDJ to 0.08 during FMA, while a correlation between bromide and aerosols is only observed during ASO ($R = 0.18$) and FMA (0.23). The positive correlation between particulate bromide and sea ice contact during ASO ($R = 0.34$) and NDJ ($R = 0.16$) confirms that the bromide observed in filter samples at NM originates from the sea ice surface. There open ocean might, however, also contribute to particulate bromide during ASO and FMA, when bromide and $\tau_{\text{water}}$ are positively correlated with PCCs of $R = 0.20$ and $R = 0.19$, respectively.

The bromide to sodium ratio from aerosol filter samples at NM does not correlate with the abundance of BrO ($|R| < 0.1$). Thus a depletion of particular bromide by heterogeneous release of reactive bromine, as observed on snow samples in the Arctic (Simpson et al., 2005) cannot be confirmed for NM. The bromide to sodium ratio shows some interesting relationships with other parameters. It is negatively correlated with temperature, solar radiation and mixing layer height during ASO, which may point towards the nature of the fractionation processes leading to an enhancement in particulate bromide. This topic requires,

however, more detailed investigations that are outside of the scope of this study.

In most cases, BrO and aerosols are positively correlated with the locally measured wind speed $v$ (during all seasons at NM, and during NDJ and FMA at AH). Together with the positive correlation between aerosols and wind speed at NM (but not at AH), this again confirms the hypothesis that higher wind speeds lead to an increase in dispersion of snow and ice particles and to a subsequent BrO release from their surfaces. One would expect that the wind speed $v_{\text{ice}}$ of the air parcels over sea ice

would even be a better indicator for particle dispersion from the frozen ocean, but this parameter is even negatively correlated to BrO and aerosols during ASO at both sites. A possible explanation for this discrepancy between local wind speed and mean wind speed over ice could be that the time the air parcels spend over sea ice, which anti-correlates with wind speed, is of more importance than a dispersion of particles. On the other hand, the wind speed over ice is strongly linked to the thickness of the boundary layer, with a PCC of up to $R = 0.8$ between $v_{\text{ice}}$ and $h_{\text{ml}}$, while both BrO and aerosols are negatively correlated to

the mixing layer height over ice. This could be caused by a stronger accumulation of BrO in a more shallow boundary layer with reduced vertical mixing, which in turn leads to a stronger release of reactive bromine owing to the non-linear nature of the bromine explosion mechanism.

BrO and aerosols are not only positively correlated with the sea ice contact time $\tau_{\text{ice}}$, but also with $\tau_{\text{land}}$, which represents the contact time with the shelf ice regions and the interior of the Antarctic ice shield - at AH only during ASO and at NM during

all seasons. This could be an indication that BrO is released from snow surfaces in the interior of the continent after deposition





of saline particles that originate from the sea ice and/or the open ocean and are transported to the inland, as previously observed far from the coast on the ice sheet of Greenland (Stutz et al., 2011). This aspect will be discussed in more detail in Section 3.4.

Except during ASO when sea ice extent is at its maximum, BrO and aerosols at NM are positively correlated to the contact time with ocean water $\tau_{\text{water}}$. This could be caused by BrO release either directly from sea spray, as observed in mid-latitudes and the tropics where BrO was found to be present in the lower ppt range at coastal stations and during ship cruises (e.g., Leser et al., 2003; Saiz-Lopez et al., 2004), or after deposition of saline particles on the snow pack, which is a likely source for the morning time peak in BrO during FMA at NM discussed in Section 3.2.

BrO and aerosols are either anti-correlated or (in case of AH during NDJ and FMA) uncorrelated to the residence time of the air parcels in the free troposphere ($\tau_{\text{free}}$), where neither sources for reactive bromine nor for particles are expected to be present.

Although one would expect that radiation would amplify the photochemical cycles that lead to the heterogeneous release of reactive bromine, BrO is anti-correlated to the integrated solar radiation along the trajectories at both sites during ASO and NDJ. This is a further indication for the importance of a heterogeneous release of $Br_2$ during the night (see Section 3.2), leading to a peak in reactive bromine right after sunrise when radiation levels are still low.

There is no clear relationship between temperature and BrO abundance at NM, with in situ temperature and mean temperature along the trajectories showing different signs. In contrast, BrO and temperature at AH are negatively correlated during ASO and NDJ both in situ and along the trajectories, a finding that is in agreement with BrO observations in the Arctic (Pöhler et al., 2010). Increased BrO release at low temperatures can be expected owing to carbonate precipitation from sea ice at low temperatures, resulting in a reduction of its buffer capacity that facilitates acidification of saline surfaces, which is an important prerequisite for heterogeneous bromine release Sander et al. (2006).

A significant impact of BrO on ozone, both from ozone soundings as columnar parameter and ozone monitor as surface parameter, is only found during ASO, with a PCC between BrO and $O_3$ of $R \approx -0.4$. There is little correlation between BrO and $O_3$ during NDJ and FMA, indicating that no significant ozone depletion takes place at other times than during polar spring, despite the fact that BrO is occasionally abundant during NDJ and FMA with VCDs of more than $1.5 \cdot 10^{13}\,\text{molec cm}^{-2}$ at NM (see Figure 7). This can be explained by a more sporadic nature of the bromine release events during summer, which are linked to stable conditions with strong temperature inversions, allowing for an accumulation of BrO without leaving sufficient time for a significant ozone depletion (Nasse, 2019).

## 3.4 BrO source-receptor analysis

Figure 12 shows source-receptor maps for BrO and aerosols at Neumayer Station during ASO (respective plots for NDJ and FMA for both stations are presented in the supplemental Figures S10 - S15). Air masses rich in BrO mainly originate from the Weddell Sea (WS) and from the coast line of the EIS east of Neumayer station. A particular hot spot is the Filchner-Ronne ice shelf (FRIS) in the south of the WS. As can be seen from the stream lines in the wind direction plot in Figure 12, air parcels are usually transported to the measurement location either from the east along the coastline of the EIS, or from the West Antarctic ice sheet over the FRIS and the WS.



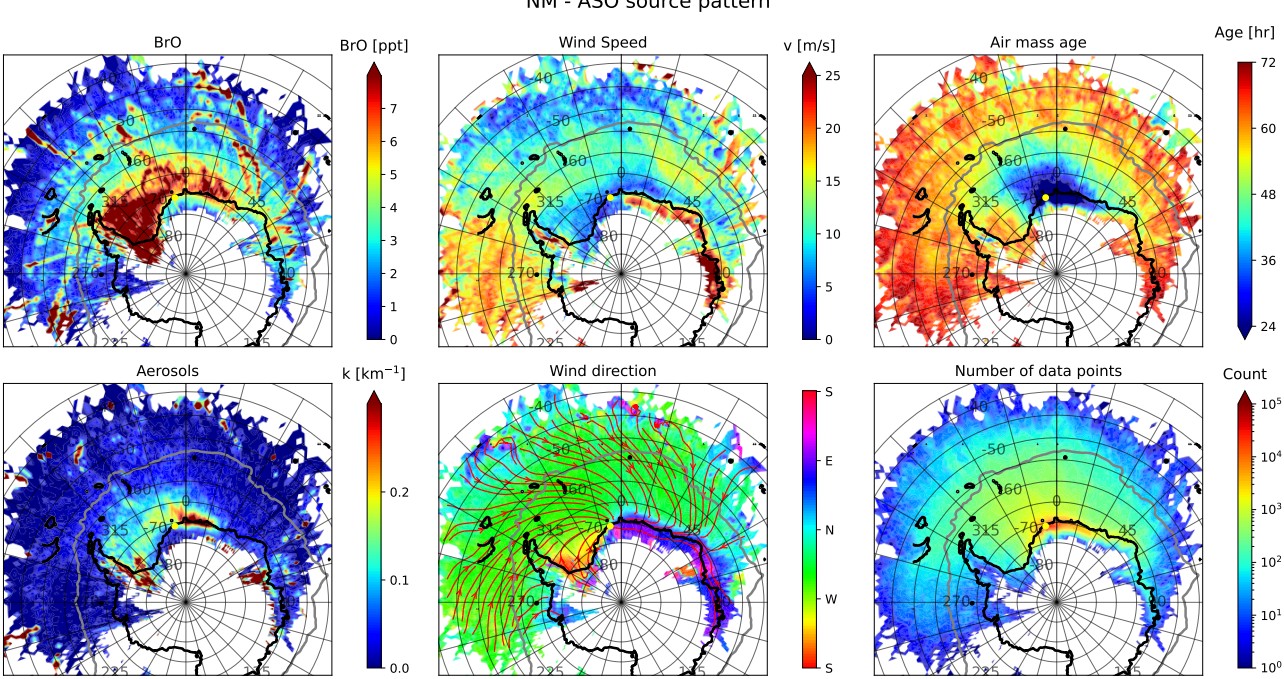

**Figure 12.** Source-receptor maps of BrO at NM during ASO, together with source-receptor maps of mean aerosol extinction, wind speed, wind direction and air mass age, as well total number of air parcels, within each $1° × 1°$ grid cell. The thick grey line indicates the mean location of the sea ice edge during ASO. The wind direction plot also contains average stream lines of the air flow (red lines with arrows). The yellow dot indicates the location of the measurement site.

The coastline east of Neumayer is an area where polynias are frequently present and where the production rate for sea ice is high (Tamura et al., 2008; Nakata et al., 2021). The newly formed sea ice is usually covered by a brine layer that represents a potential source for reactive bromine (Morin et al., 2008). Frost flowers formed on top of the brine layer, which represent fragile crystals that easily become airborne, contribute with 40% to the deposition of sea salt on the ice shelf and continental ice sheet (Kaspari et al., 2005). Figure 12 shows that the aerosol extinction is significantly increased if the air comes from this

area, which means that particles, probably with high salinity, are transported from there to NM. The eastern boundary of this area with enhanced extinction is located at about $20°$E, where a persistent area of open water with a high sea ice production rate named the Cape Darnley Polynya is located (Nihashi and Ohshima, 2015).

The fact that air masses originating from the FRIS are high in BrO is surprising since no ocean water as the primary source for reactive halogens is present in this area. A possible explanation would be that dense air masses are transported from the

interior of the continent, pass the FRIS and are afterwards in contact with sea ice in the WS. However, satellite measurements also detect high amounts of BrO in the FRIS region (e.g., Theys et al., 2011; Schönhardt et al., 2012), confirming that this area represents a hot spot for the production of reactive bromine. This requires a sufficient supply of halides from the ocean



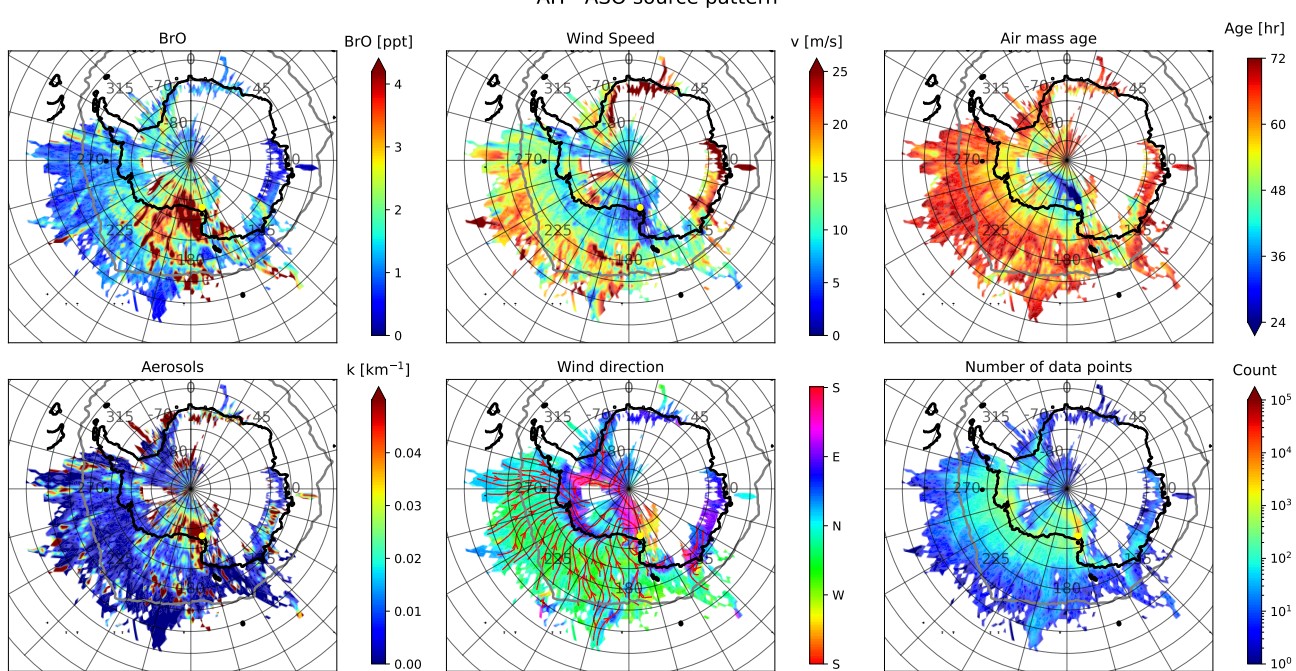

**Figure 13.** Same as Figure 12, but for AH. Note the different colour scales for BrO and aerosols.

and/or the sea ice to this region via deposition of saline particles. Possibly, cyclonic activity leads to the transport of air masses from the southern WS, which represents a region of high sea ice production rate (Tamura et al., 2008; Nakata et al., 2021;

Nihashi and Ohshima, 2015), to the ice shelf. However, the interplay between transport of saline particles to the FRIS and the subsequent release of reactive bromine in the presence katabatic winds, which lead to the transport of BrO-rich air masses to NM, remains a subject for future investigations that require a more detailed analysis of the meteorological conditions in this area.

The source-receptor analysis for AH presented in Figure 13 shows that air masses observed at this site mainly originate from

the coastal region of West Antarctica, the Ross Sea (RS), the Amundsen Sea (AS), as well as the Bellingshausen Sea (BS), and to a smaller extent also from Wilkes Land (WL) to the west of the RIS. The average flow pattern shows that air masses are frequently transported south-eastwards over the sea ice, before they turn westwards and move along the coast of West Antarctica, then move polewards over the RIS, become deflected by the Transantarctic Mountains and finally approach AH as southerly winds. During ASO, BrO is mainly increased if the air masses originate from the sea ice over the RS and from the

RIS. Aerosols are much lower (by about a factor of 10) at AH than at NM. Air masses travelling over the RIS in the vicinity of AH are subject to increased aerosol load, which probably originates from the large polynia usually present east of Ross Island that potentially supplies the surface of the shelf ice with halides. Another region of increased aerosol production is the Terra Nova Bay (TNB) north of AH, where sea ice production rates are high as well (Nihashi and Ohshima, 2015).



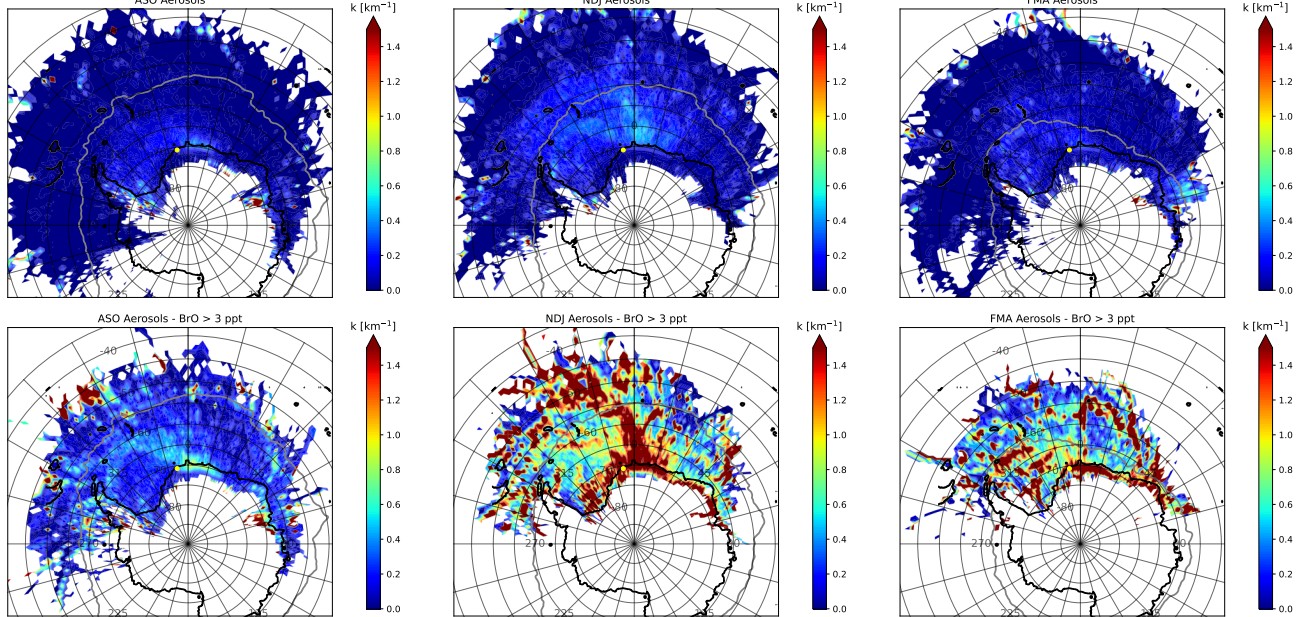

**Figure 14.** Source-receptor distribution of mean aerosol extinction at NM for all data (top row) and only for BrO > 3 ppt (bottom row) during ASO (left column), NDJ (middle column) and FMA (right column). Note the different colour scale compared to Figures 12 and 13.

The air masses arriving at AH occasionally originate from the WS and RIS and traverse the Antarctic ice sheet, some of which are high in BrO. These cases of transcontinental transport of reactive bromine will be discussed in Section 3.5.

The impact of aerosols on the abundance of BrO is illustrated in Figures 14 and 15, where source-receptor maps of aerosols for all measurements (top panels) are compared to those when BrO VMR > 3 ppt (bottom panels) during each season. During ASO, there is little enhancement of aerosols when BrO is high, indicating that the release of reactive bromine from aerosols does not play a dominant role during this season. The only exception is the Amery Ice Shelf (AIS) located in West Antarctica, which appears to be a strong source of aerosols. The situation is very different during NDJ and FMA, when aerosols are often strongly enhanced in the presence of BrO. For NM, this is in particular the case when air masses either originate from the RIS, from the sea ice north of NM, or from the coastline of the EIS east of the measurement site. Air coming from the EIS is particularly high in aerosols when BrO is high during FMA, when new sea ice forms along the coast that represents a potential source for reactive bromine. At AH, air masses containing enhanced BrO during NDJ and FMA mainly originate from the coastal areas of the RS and the AS, as well as from the RIS and the TNB. These often contain enhanced aerosols, but compared to NM there are no clear patterns apparent in the aerosol source-receptor maps.

## 3.5 Transcontinental transport of reactive bromine

As can be seen from Figure 13, some of the air parcels arriving at AH originate from the WS and even from the EIS in the vicinity of NM. They usually travel across the continent past the South Pole and along the Transantarctic Mountains before




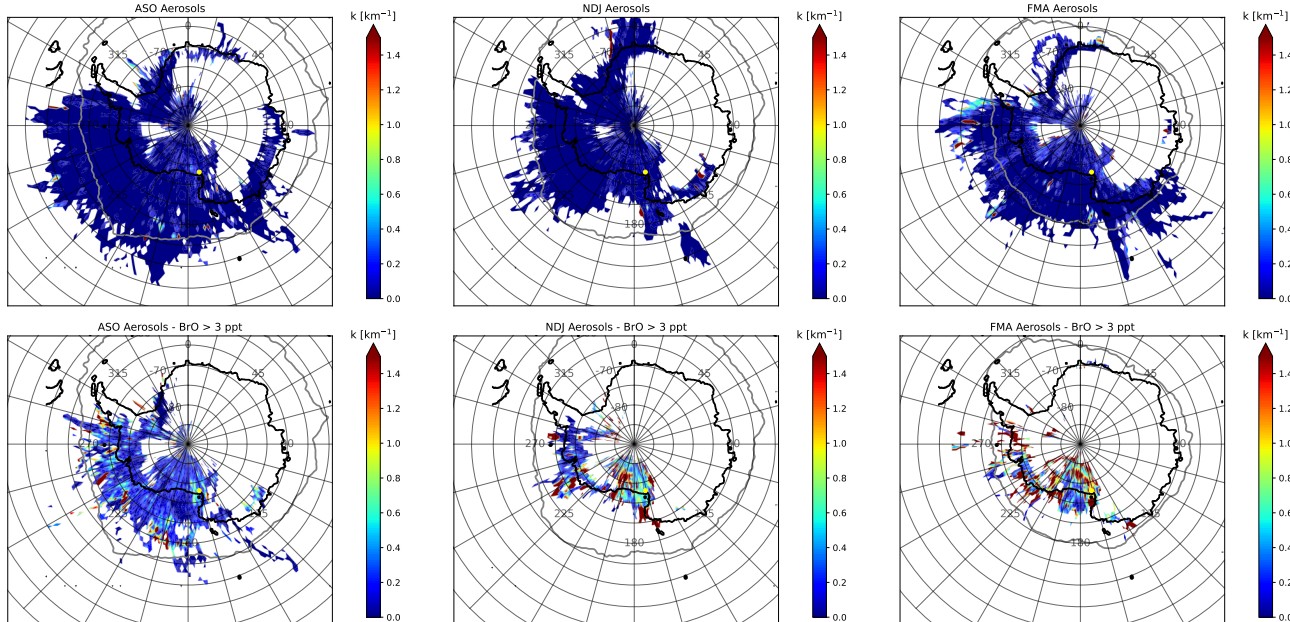

**Figure 15.** Same as Figure 14, but for AH.

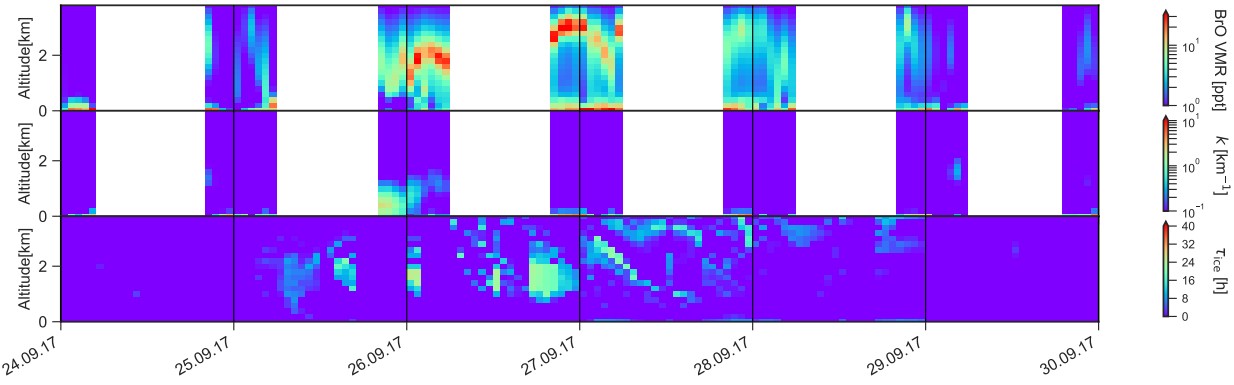

**Figure 16.** BrO VMR profiles (top), aerosol extinction (centre) and sea ice contact time (bottom) observed at AH between 24.9. and 30.9.2017.





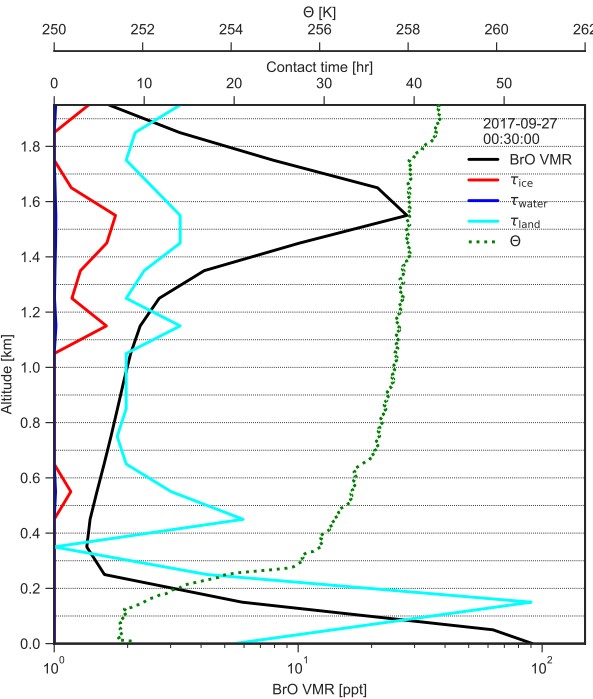

**Figure 17.** Vertical profile of BrO VMR, as well as contact times with sea ice, land and water at AH on 27. September 2017, 00:30, together with the vertical profile of potential temperature Θ inferred from a coincident balloon sonde measurement. The dotted horizontal lines indicate the MAX-DOAS retrieval grid.

they finally reach AH. In the available data set (2012 - 2021), a transcontinental transport of air masses with enhanced BrO (VMR > 3 ppt) was observed on 24 days (1.8% of all days with enhanced BrO). As an example, Figure 16 shows BrO vertical profiles and sea ice contact time for a period of six days during late September 2017. Enhanced BrO with VMR > 20 ppt is observed at altitudes above 1.5 km from 26. to 28. September. An additional surface layer with high mixing ratios (VMR > 100 ppt) is present at altitudes below 200 m from 27. September on. As can be seen from the temperature profile in Figure 17,

the BrO surface layer resides within the boundary layer that is confined by a strong temperature inversion at ≈ 280 m altitude, while the uplifted layer resides in the free troposphere. The respective backward trajectories shown in Figure 18 reveal that the air masses of the uplifted BrO layer originate from the WS, where they had contact with sea ice while they resides in the mixing layer. The respective air parcels travelled over the continental ice sheet for a duration of more than 60 h before reaching the measurement site, during which they had no contact with any potential source for reactive bromine.

A residual inversion layer is present right above the uplifted layer of BrO at 1.8 km altitude (see Figure 17), most probably caused by advection of the boundary layer that initially resided in the southern WS where BrO was emitted. The fact that high concentrations of reactive bromine are sustained while the air parcels travel over the continental ice sheet for about three days is surprising, in particular since the MAX-DOAS instrument detects no significant aerosol extinction at the height of the BrO




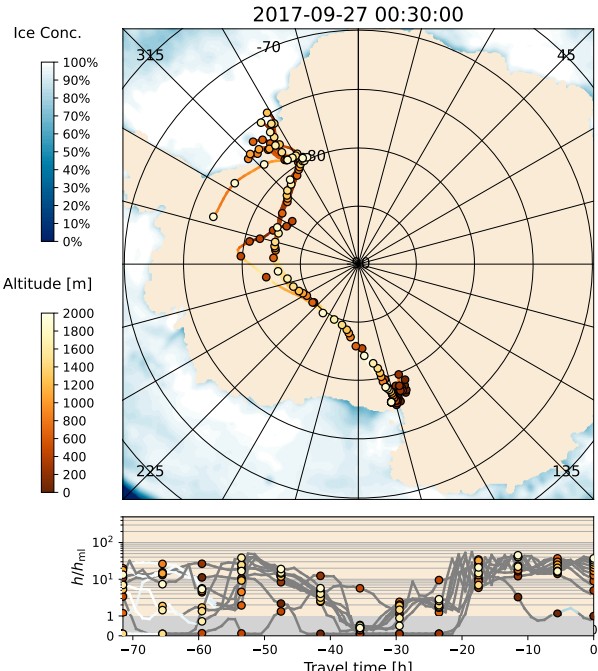

**Figure 18.** Three days back trajectories ending at AH on 27.9.2017, 00:30 UTC. Trajectories are shown for every second profile layer only. For further description see caption of Figure 3.

layer, which could sustain BrO concentrations by recycling on their surfaces. Model calculations indicate that, in the absence
of sources, most BrO should be converted to reservoir species during their travelling time over the continent (Piot and von Glasow, 2008; Yang et al., 2005). Thus either the initial BrO concentration in the WS was extraordinarily high, or reactive bromine was recycled on the surfaces of very small saline particles not detectable by MAX-DOAS.

The near-surface air masses with enhanced BrO had contact with the RIS in the vicinity of AH until about 20 h before arrival at the measurement site before they became uplifted above the mixing layer. This shows that the shelf ice surface acts as a
source for reactive bromine, and that high levels of BrO (> 100 ppt) can sustain over a period of almost one day without any contact surfaces on which bromine recycling could occur.

## 4   Summary and Conclusions

In this study, the dynamics chemistry of reactive bromine in the Antarctic boundary layer is investigated using MAX-DOAS measurements from two coastal stations, covering about 20 years of measurements at Neumayer Station, and 10 years at Arrival
Heights. The resulting data set of more than $100\,000$ vertical profile pairs of BrO VMR and aerosol extinction is compared to



a variety of co-located measurements of meteorological and chemical parameters of the atmosphere. Furthermore, sources of reactive bromine are investigated using a source-receptor analysis based back-trajectories together with sea ice maps.

In agreement with previous ground-based and satellite borne measurements, highest BrO VCDs are observed after polar sunrise in August/September. The period during which enhanced BrO is present lasts until October at AH and early November

at NM. There are, however, a significant number of observations of enhanced BrO also during summer and autumn, in particular at NM where BrO VCDs frequently exceed $1 \cdot 10^{13}$ molec cm$^{-2}$. Such amounts of BrO are difficult to detect by satellite owing to a lack of accuracy in the separation of the stratospheric and tropospheric partial columns of BrO and the shorter light path for nadir observations than for ground-based off-axis measurements (Seo et al., 2019), while the detection limit (median retrieval error) of the tropospheric BrO VCD from our ground-based MAX-DOAS measurements amounts to only

$0.3 \cdot 10^{13}$ molec cm$^{-2}$ and the MAX-DOAS technique eliminates the stratospheric contribution to the BrO signal intrinsically.

The majority of BrO events during spring have their maximum VMR near the surface, but the probability for the occurrence of uplifted BrO profiles increases significantly in summer and autumn, when higher temperatures and solar irradiance lead to enhanced vertical mixing. The reduced BrO concentrations observed near the ground during ASO and NDJ indicate that the snow surface might be a sink during the day owing to insufficient bromide concentrations that could lead to an amplification

of the reactive bromine concentration in the gas phase. These uplifted profiles, which are more frequent at AH than at NM owing to topographically induced vertical mixing at AH, represent a potential pathway of BrO to the free troposphere, where reactive bromine is expected to have a significant impact on the oxidative capacity of the atmosphere and eventually on climate by affecting the ozone budget and the formation of clouds (von Glasow et al., 2002b, a).

The diurnal variation of BrO at NM shows a strong asymmetry between am and pm, with significantly higher values during

the morning. This is most likely caused by the nocturnal release of Br$_2$ from the local snow surface by heterogeneous reaction with ozone, and the subsequent photolysis of Br$_2$ during sunrise. Extremely high peaks in BrO VCD right after sunrise are observed during autumn, when sea ice extent is at its minimum and local release of reactive bromine dominates. Here, deposition of sea salt aerosols dispersed from the open ocean on the ice shelf probably plays an important role as a supply for halides, as can be seen from the positive correlation between BrO and sea water contact time during summer and autumn at NM. In

contrast, springtime BrO events are much more widespread, when air masses rich in reactive bromine transported from the frozen ocean to the measurement site leave more processing time for the depletion of ozone.

A regression analysis based on a variety of meteorological parameters, as well as the residence time over sea ice, open water and the continental ice sheet, shows a strong correlation between BrO and aerosol abundance during all seasons. This supports the hypothesis that aerosols play an important role in the release of reactive bromine in polar regions. A positive correlation

between particulate bromide and gaseous BrO during ASO confirms a direct release of bromine radicals from aerosol surfaces. However, saline particles deposited to the snow surface most probably lead to an accumulation of bromide in the snowpack, where the bromine explosion mechanism can be triggered at a later time. This hypothesis is supported by the fact that, during ASO, the BrO VMR is positively correlated to the contact time with the continental ice sheet and ice shelf regions. Our source-receptor analysis reveals that the FRIS is one of the main sources of air containing high BrO concentrations at NM.



At NM, where air mainly originates from the ocean and coastal areas, BrO is positively correlated with the contact time with open water, with increasing correlation coefficients during summer and spring as the sea ice retreats. This indicates that significant amounts of reactive bromine released the open ocean serves as a source for reactive bromine, with the bromine being released from sea spray and/or sea salt aerosols, either while the particles are airborne or after their deposition on the snow surface. In contrast, there is a much stronger influence of continental air at AH, and BrO enhancements are less frequent

and weaker than at NM. The source-receptor analysis reveals that at AH air rich in BrO mainly comes from the sea ice and the RIS. There are, however, some cases when air enhanced in BrO travelled from the FRIS at the adjacent side of the continent over the Antarctic ice shield to AH, showing that reactive BrO can be sustained aloft for several days in the absence of any sources.

*Code and data availability.* Time series of vertical profiles of BrO and aerosols retrieved from the MAX-DOAS instruments at Neumayer
and Arrival Heights are available on the Pangaea data server (Frieß, 2022a, b). The DOASIS analysis software and the HEIPRO retrieval algorithm and the DOASIS analysis software are available by Udo Frieß upon request.

*Author contributions.* UF designed and built the instruments, has been responsible for the development of the HEIPRO algorithm, and conducted the data analysis and interpretation. UP supported the design of the instrument and the interpretation of the data since they have been operational. KK, RQ and DS have been responsible for the operation of the MAX-DOAS instrument at Arrival Heights, designed parts
of the entrance optics, and provided ancillary data from AH. HS is responsible for the meteorological observations at NM and provided meteorological data. RW is responsible for the trace gas measurements at Neumayer Station including the MAX-DOAS instrument. He performed the chemical analysis of the aerosol filter measurements and provided ancillary data from NM.

*Competing interests.* The authors declare that they have no competing interests.

*Acknowledgements.* Our sincerest thanks go to the countless scientists and technicians at Neumayer Station and Scott Base, who operated
and maintained the MAX-DOAS instrument over so many years. Their immense support made sure that the instruments provided continuous measurements of high quality. Would would also like to thank Paul Johnston, Alan Thomas and Steven Wood from NIWA for their support of the operation of the AH MAX-DOAS instrument. The authors appreciate the support of the Automatic Weather Station Program and Antarctic Meteorological Research Center for the McMurdo radiosonde data (Matthew Lazzara, NSF grants number ARC-0713843, ANT-0944018, and ANT-1141908). We thank the National Oceanic and Atmospheric Administration (NOAA) for providing AH surface ozone
data, and for making GDAS meteorological fields as well as the HYSPLIT trajectory model available. We would also express our gratitude to the EUMETSAT Data Center (EDC) for providing OSI-SAF sea ice products. Many thanks to Alexei Rozanov for providing the SCIATRAN radiative transfer model, which serves as the forward model for the HEIPRO algorithm. Parts of this study were funded by the Deutsche Forschungsgemeinschaft (DFG), grant numbers FR2497/2-1, FR2497/3-1, and FR2497/3-2.



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
