# Peer review of "Source Mechanisms and transport Patterns of tropospheric BrO: Findings from long-term MAX-DOAS Measurements at two Antarctic Stations"

_EGUsphere, 2022_

## Referee Comment (RC1)

*Review of Frieβ et al. (2022): Source Mechanisms and transport Patterns of tropospheric BrO: Findings from long-term MAX-DOAS Measurements at two Antarctic Stations*

The manuscript presents 19 and 10 years of MAX-DOAS observations of tropospheric BrO and aerosols from the Antarctic stations of Neumayer (NM, 70º 40' S, 8º 16' W) and Arrival Heights (AH, 77º 49' S, 166º 39' E), respectively. Using this extensive data series, the authors investigate linear correlations between numerous parameters such BrO at surface, aerosol extinction, wind speed, contact time with sea ice, etc. After investigating these regressions, the authors conclude that (1) there is a correlation between BrO and aerosol extinction indicating that airborne saline particles are a dominant source of bromine; (2) there is a correlation between BrO and air masses' contact time with sea ice and with ice sheet, indicating that not only the sea ice but also the snow over the ice sheet is a source of bromine; (3) when the sea ice retreats, the ocean is also a source of reactive bromine. Furthermore, the authors suggest possible source regions of reactive bromine for NM and for AH sites. In addition, the authors point out the detection of an early morning peak of tropospheric BrO and also of sustained uplifted plumes of BrO crossing the continent for several days.

Throughout the manuscript, the authors discuss sources and processes of tropospheric reactive bromine in the Antarctic, addressing relevant scientific questions. Furthermore, to the reviewer' knowledge, this is probably the largest dataset of BrO observations in the pristine Antarctic troposphere. Chemical models and the scientific community can indeed benefit from these observations.

The manuscript is well presented and the methodology is well described. Overall, publication is recommended after addressing the following comments.

**General Comments:**

- Although the number of variables studied in the regression analysis is outstanding, the conclusions gathered after those regressions seem sometimes overstated. For instance, a correlation of -0.13 (BrO-radiation) is stated as "anti-correlated" and a conclusion is obtained after. However, a R= -0.13 indicates a rather low (or even negligible) linear relationship between two variables since knowing variable A would explain only 1.69% of the variance of variable B (i.e., close to negligible correlation). Similar applies to BrO-aerosols at AH (R=0.28) or BrO-$\tau$ice at AH (R=0.17), etc. A revision of statements like "this provides evidence", "this confirms" etc throughout the manuscript is suggested along with the conclusions gathered based on not high (or even very low) R. Also, all this study is based on linear regression between pair of variables. Given the complex nature of the system and the huge amount of data and variables studied, could a multivariate regression provide additional information?
- Throughout the manuscript, some context of the obtained values of BrO and aerosols is missing (i.e., how the values at NM and AH do compare with previous published works in Antarctica and/or the Arctic region?).
- The manuscript may benefit if providing the corresponding BrO VMR (i.e., ppt) along with VCD (e.g., detection limit of BrO, P16, L342). Also including BrO ppt and aerosol ranges observed throughout all the years could provide a nice overview of the data.
- In the manuscript, aerosol extinction coefficients are discussed at both sites based on MAX-DOAS observations. Is there any cloud filter applied when retrieving those aerosols? If no cloud filter is applied, the obtained "aerosol" results might well be due to clouds instead. This limitation should be mentioned in the manuscript so the conclusions reached after the "aerosol" data could be taken with caution.
- Please, discuss the possible influence of stratospheric BrO on the observations at high SZA (see also specific comments to this regard, e.g., P9, L220-224).

**Specific Comments:**

**P1, L19:** "BrO can be sustained for several days". A range deliming "several" could assist the conclusion (e.g., ~2 days).

**P2, L31:** missing the reference to the review work of Angot et al. 2016.

**P3, L57:** also for CCM (e.g., Fernandez et al., 2019).

**P5, L111:** missing the reference to the 25-year climatology of Silva et al., 2022.

**P6, L131-138:** What is the FOV of each instrument? Why 0º is not included in the scan? Is the negative elevation angle used in the retrieval (also P10, L235)?

**P7, L175-177:** Does this affect to any of the conclusions reached in the manuscript?

**P8, L196-198:** "Weighted by the time difference", please describe a bit better this approach.

**P8, L208:** To avoid self-referencing, could only one of the 4 references to HEIPRO be used?

**P9, L220-224:** "all dSCDs measured during one hour serve as input for a single retrieval". For low SZA, this might be a good approach. However, for high SZA this most probably provide results including stratospheric information since a single a priori profile (and AMF) is assumed constant during that hour. If the measurement vector consists on observations at each given angle during 1 h, wouldn't the stratospheric information change a lot during 1h at high SZA? How could this affect the authors' results at high SZA (e.g., the BrO peak the authors observe after sunrise and before sunset; P20, L1)?

**P11, L278:** Why is the residence time weighted?

**P13, figure 4:** Please, include the error bars in the figure. Also, why are there 4 different BrO profiles retrieved for one single data and time?

**P15, last paragraph:** Since on 7-9[th] October there was very high aerosol extinction coefficient, how reliable are the BrO VMR values retrieved on those days?

**P16, figure 5:** Since e.g. the pressure does not seems to be addressed, for the sake of simplicity, that label and graph could be removed.

**P17, L363:** This section presents the spatio-temporal variability of BrO. Since aerosols are also addressed throughout the manuscript, a similar section for aerosols would benefit the draft.

**P17, L364:** Does BrO VCD refer to total column VCD or only tropospheric column VCD? Please, clarify in the text. Also, since as stated by the authors the MAX-DOAS observations are sensitive to the first 1.5 km (P9, L232), should not the VCD refer only to those km?

**P18, L364:** By the different meteorological conditions and also by the location of each research site.

**P19, L390:** Please, state the meaning of having higher bromide-to-sodium ratio.

**P19, L401:** How does the surface BrO daily evolution compare to the work of Nasse et al. 2019 where, with a LP-DOAS, they detected up to 60ppt around noon?

**P21, L448:** Regarding BrO getting into the free troposphere, what is the evolution of the boundary layer height?

**P22, figure 10:** Is this figure considering only BrO data above detection limit?

**P22, L56-457:** p-values close to 0 indicate the *significance* of the R obtained.

**P23, figure 11:** Which BrO and aerosol from MAX-DOAS are used? Surface values? (also in P24, L:461)

**P25, L523:** The source-receptor analysis seems very clarifying. Please detailed a bit better the procedure behind this sort of analysis (e.g., which altitude of the BrO profile is used for this analysis)?

**P25, L527:** Do observations at Belgrano or Halley support the FRIS as a BrO hot spot? Also P26, L542; P27, L547.

**P27, L557-558:** Do observations at Dumont d'Urville support the TNB as an aerosol hot spot?

**P30, figure 17:** The time referred to is 00:30. Given the high SZA, how can the stratospheric influence be excluded?

**P30, L586-588:** Can a stratospheric intrusion be ruled out? Are O3 values available?

**P32, L610:** Please provide the BrO VCD also in ppt (easier to compare with other ground-based observations).

**P32, L613-615:** (Snow surface as a BrO sink during the day). This may contradict with e.g. the work of Saiz-Lopez et al. 2007 with a LP-DOAS. Same applies with the observed diurnal variation at NM with an early morning peak.

**Technical Corrections:**

**P1, L1:** Polar Regions shall be polar regions (no block letters).

**P5, L109:** Katabatic shall katabatic (no block letter).

**P15, L319:** 28. August shall be 28$^{th}$ August. This happens to all dates throughout the manuscript.

**P31, L591:** A "," is missing after Thus

**P31, L598:** The "dynamics chemistry"? do the authors refer to the chemistry and the dynamics?

---

## Author Response (AR1)

We thank both anonymous reviewers for their constructive comments, which certainly help to improve the overall quality of our manuscript. In the following, we reply to the comments point by point. Comments by the referees will be shown in **bold font**, our replies to the comments in normal font, and changes made to the manuscript in *italic font*. Page and line numbers refer to the first version of the manuscript.

**REPLIES TO ANONYMOUS REFEREE #1**

**The manuscript presents 19 and 10 years of MAX-DOAS observations of tropospheric BrO and aerosols from the Antarctic stations of Neumayer (NM, 70° 40' S, 8° 16' W) and Arrival Heights (AH, 77° 49'S, 166° 39' E), respectively. Using this extensive data series, the authors investigate linear correlations between numerous parameters such BrO at surface, aerosol extinction, wind speed, contact time with sea ice, etc. After investigating these regressions, the authors conclude that (1) there is a correlation between BrO and aerosol extinction indicating that airborne saline particles are a dominant source of bromine; (2) there is a correlation between BrO and air masses' contact time with sea ice and with ice sheet, indicating that not only the sea ice but also the snow over the ice sheet is a source of bromine; (3) when the sea ice retreats, the ocean is also a source of reactive bromine. Furthermore, the authors suggest possible source regions of reactive bromine for NM and for AH sites. In addition, the authors point out the detection of an early morning peak of tropospheric BrO and also of sustained uplifted plumes of BrO crossing the continent for several days.**
**Throughout the manuscript, the authors discuss sources and processes of tropospheric reactive bromine in the Antarctic, addressing relevant scientific questions. Furthermore, to the reviewer' knowledge, this is probably the largest dataset of BrO observations in the pristine Antarctic troposphere. Chemical models and the scientific community can indeed benefit from these observations.**
**The manuscript is well presented and the methodology is well described. Overall, publication is recommended after addressing the following comments.**

**General Comments:**
**• Although the number of variables studied in the regression analysis is outstanding, the conclusions gathered after those regressions seem sometimes overstated. For instance, a correlation of -0.13 (BrO-radiation) is stated as "anti-correlated" and a conclusion is obtained after. However, a R= -0.13 indicates a rather low (or even negligible) linear relationship between two variables since knowing variable A would explain only 1.69% of the variance of variable B (i.e., close to negligible correlation). Similar applies to BrO-aerosols at AH (R=0.28) or BrO-ice at AH (R=0.17), etc. A revision of statements like "this provides evidence", "this confirms" etc throughout the manuscript is suggested along with the conclusions gathered based on not high (or even very low) R. Also, all this study is based on linear regression between pair of variables. Given the complex nature of the system and the huge amount of data and variables studied, could a multivariate regression provide additional information?**
Given the complex nature of bromine chemistry, a linear relation between and thus high correlation coefficients between different parameters, can certainly not be expected. However, the p-values are very close to zero for most pairs of variables, including BrO-aerosols and also BrO-radiation (see supplemental Figure S9), which means that the is a very small probability that these variables are statistically independent. A correlation coefficient of -0.13 between BrO and radiation is indeed low, and what we meant to say is that it is surprising to see that there is not positive correlation between both variables, which would be expected from simple photochemistry since the bromine explosion cycle requires the photolysis of $Br_2$. We have therefore changed the wording as follows (P25, L506):
*Although one would expect that radiation would amplify the photochemical cycles that lead to the heterogeneous release of reactive bromine, leading to a positive correlation between BrO and integrated solar radiation along the trajectories, we find instead a small negative PCC between these variables at both sites during ASO and NDJ.*

Apart from the simple linear correlation analysis we have chosen for our study, a multivariate analysis could also be applied for our data sets, or other more sophisticated approaches such as principal component analysis. This would, however, not affect the regression coefficients, which are independent of the regression method.

**• Throughout the manuscript, some context of the obtained values of BrO and aerosols is missing (i.e., how the values at NM and AH do compare with previous published works in Antarctica and/or the Arctic region?).**
We find it difficult to directly compare BrO abundances at different locations and different times, in particular since most other studies focus on shorter time scales than in the present study. It is also not our intention to provide a full review of existing BrO measurements, which can be found elsewhere. We have added the following statement to the introduction:
*The abundance of BrO varies substantially with time and location, with maximum mixing ratios reported in different studies ranging from 5 ppt over the sea ice near Scott Base (Zielcke,2015) over 50 ppt from ship-borne measurements over the frozen Weddell Sea (Wagner et al., 2007) to 111 ppt at Neumayer Station using a long-path DOAS instrument that directly measures the BrO mixing ratio near the surface (Nasse, 2019). For a detailed review of halogen observations in polar regions, see e.g. Simpson et al. (2015).*

**• The manuscript may benefit if providing the corresponding BrO VMR (i.e., ppt) along with VCD (e.g., detection limit of BrO, P16, L342). Also including BrO ppt and aerosol ranges observed throughout all the years could provide a nice overview of the data.**
We replaced "*no BrO is detected above the detection limit*" in L342 with "*only very small BrO VMR below 1 ppt were detected*". We have added two figures with the multi-annual variation of the ASO BrO VMR, aerosol extinction and duration of sea ice contact to the supplement (Figures S14 and S15). A discussion and interpretation of the multi-annual BrO trend is, however, beyond the scope of this study since this requires a detailed investigation of the inter-annual variation of the driving parameters of BrO release.

**• In the manuscript, aerosol extinction coefficients are discussed at both sites based on MAX-DOAS observations. Is there any cloud filter applied when retrieving those aerosols? If no cloud filter is applied, the obtained "aerosol" results might well be due to clouds instead. This limitation should be mentioned in the manuscript so the conclusions reached after the "aerosol" data could be taken with caution.**
This is an important point that we missed to mention in the manuscript. Indeed, no cloud filter is applied, because clouds and blowing snow cannot be distinguished by the retrieval. We would therefore not be able to investigate the impact of blowing snow on BrO if applying a cloud filter. We have therefore added the following statement to Section 2.5:
*Note that a cloud filter has not been applied to the retrieved profiles, because clouds and blowing snow cannot be distinguished unambiguously, and it would not be possible to investigate the impact of blowing snow on BrO when applying a could filter. Therefore parts of the AOD reported in this study are owing to extinction by clouds and not aerosols or snow particles.*

**• Please, discuss the possible influence of stratospheric BrO on the observations at high SZA (see also specific comments to this regard, e.g., P9, L220-224).**
See our replies to the specific comments below.

**Specific Comments:**
**P1, L19: "BrO can be sustained for several days". A range deliming "several" could assist the conclusion (e.g., ~2 days).**

The statement "*BrO can be sustained for several days*" has been replaced with "*BrO can be sustained for at least three days*".

**P2, L31: missing the reference to the review work of Angot et al. 2016.**
**P3, L57: also for CCM (e.g., Fernandez et al., 2019).**
Thank you for pointing this out, the according references were added.

**P5, L111: missing the reference to the 25-year climatology of Silva et al., 2022.**
Unfortunately, we are not aware of a publication of Silva et al. related to the meteorology at Neumayer.

**P6, L131-138: What is the FOV of each instrument?**
In L129, "*Scattered sunlight is collected by telescope units and fed to the spectrometer units using quartz fibre bundles.*" Is replaced by "*Scattered sunlight is collected by telescope units with a field of view of ~0.5° and fed to the spectrometer units using quartz fibre bundles.*"

**Why 0º is not included in the scan? Is the negative elevation angle used in the retrieval (also P10, L235)?**
Measurements at an elevation angle of 0° are not included because these cannot be attributed unambiguously to light scattered either from the atmosphere or from the ground, and it is not trivial to simulate this geometry with our radiative transfer model. Furthermore, 0° measurements are very sensitive to small misalignments, varying sensitivity of the instrument over the field of view, atmospheric effects, such as light refraction (mirages) during strong inversions. The negative angles are of course included in the retrieval since, as discussed in the manuscript, these strongly enhance the sensitivity, in particular below the measurement altitude in case of Arrival Heights that is located at an elevated site. The following sentence has been added in L211:
*Note that all viewing directions listed in Section 2.2, including negative elevation angles, are included in the retrieval.*

**P7, L175-177: Does this affect to any of the conclusions reached in the manuscript?**
Yes, as stated in this paragraph, the fact that parts of the BrO detected on the filter samples originates from the gas phase and not from particles means that an unknown fraction of the bromine considered a particulate is actually not from aerosols.

**P8, L196-198: "Weighted by the time difference", please describe a bit better this approach.**
The following statement has been added at L198:
*This is done by setting the intensity of the Fraunhofer reference at a certain time t as $I_{ref}(t) = I_1 + (I_2 - I_1) * (t-t_1)/(t_2 - t_1)$ with $I_1$ and $I_2$ being the intensity of the zenith sky spectra measured prior to and after the off-axis measurement at times $t_1$ and $t_2$, respectively.*

**P8, L208: To avoid self-referencing, could only one of the 4 references to HEIPRO be used?**
We feel that it is appropriate to cite the publication on the theoretical basis of HEIPRO (Frieß et al, 2006) as well as on application to BrO measurements in polar regions (Frieß et al, 2011), but we removed the references to publications where HEIPRO is part of intercomparison exercises (Frieß et al., 2016, 2019).

**P9, L220-224: "all dSCDs measured during one hour serve as input for a single retrieval". For low SZA, this might be a good approach. However, for high SZA this most probably provide results including stratospheric information since a single a priori profile (and AMF) is assumed constant during that hour. If the measurement vector consists on observations at each given angle during 1 h, wouldn't the stratospheric information change a lot during 1h at high SZA? How could this affect the authors' results at high SZA (e.g., the BrO peak the authors observe after sunrise and before sunset; P20, L1)?**

As explained in Section 2.4, the DOAS fit is performed using a Fraunhofer reference from each individual elevation scan, which has a duration of approximately 15 minutes and not an hour. The influence of stratospheric BrO is further minimised by using a Fraunhofer reference interpolated in time. The stratospheric contribution to the measurements should therefore be very small, but is probably not completely negligible at high SZA. We are not aware of any publication discussing quantitatively the impact of stratospheric BrO on tropospheric retrievals, and an assessment of these effects is beyond the scope of this study. The box-AMFs (Jacobians) are simulated for the SZA (and SAA) of each individual measurement and the variation of the light path during one hour of measurements is accounted for. However, the assumption that the tropospheric profile remains constant during one hour is a strong simplification that was necessary because otherwise we would not be able to handle the computational effort given the large amount of data. The retrieval of all profiles took several months.

**P11, L278: Why is the residence time weighted?**
The weighting makes sure that the residence time over ice is zero if there is no ice ($\rho_{ice} = 0$), and equal to the total residence time of the air parcel if the location is completely covered by ice ($\rho_{ice} = 1$). The same applies to the residence time over water, with $\rho_{water} = 1 - \rho_{ice}$.

**P13, figure 4: Please, include the error bars in the figure. Also, why are there 4 different BrO profiles retrieved for one single data and time?**
As explained in the caption and denoted in the legend of Figure 4, only the black line represents the BrO profile, while the other lines show the contact times over ice, water and land. To avoid any misunderstanding, the figure caption has been modified as follows:
*Example for a vertical profile of BrO at NM (black line, bottom axis) together with the contact times (top axis) of the air parcels over ice (red), open water(blue) and land (cyan) on 10 September 2016, 09:30 UTC. The grey shaded area shows the random error of the BrO profile. Dotted horizontal lines indicate the retrieval layer boundaries.*

**P15, last paragraph: Since on 7-9th October there was very high aerosol extinction coefficient, how reliable are the BrO VMR values retrieved on those days?**
Indeed, the averaging kernels show that there is little sensitivity above 1 km altitude. We feel that this has already been addressed in the manuscript by stating that "…*uplifted BrO layers with peak altitudes around 1 km and BrO VMRs above 10 ppt are retrieved, despite the fact that the averaging kernels indicate a very low sensitivity to these altitudes owing to the low visibility that results in large fluctuations of the BrO vertical profiles.*"

**P16, figure 5: Since e.g. the pressure does not seems to be addressed, for the sake of simplicity, that label and graph could be removed.**
We would like to keep the pressure time series in the graph in order to provide an overall picture over the meteorological conditions during the observations, while the readability of the figure would not be strongly improved if the pressure would be removed.

**P17, L363: This section presents the spatio-temporal variability of BrO. Since aerosols are also addressed throughout the manuscript, a similar section for aerosols would benefit the draft.**
*The seasonal variation of aerosols was added to Figure 7. There is, however, no clear relationship between aerosols and BrO on a seasonal basis. The following statement was added at L386:*
*In contrast to short-term variations in BrO and aerosols discussed below, there is no clear relationship between the BrO VCD and the AOD apparent in Figure 7. At NM, aerosols are present at equal amounts over the year, while highest amounts of aerosols are present during the period of lowest BrO abundance in FMA.*

**P17, L364: Does BrO VCD refer to total column VCD or only tropospheric column VCD? Please, clarify in the text. Also, since as stated by the authors the MAX-DOAS observations are sensitive to the first 1.5 km (P9, L232), should not the VCD refer only to those km?**

Throughout the paper, the terms VCD and AOD refer to the tropospheric column since MAX-DOAS measurements are not sensitive to high altitudes, as discussed in detail in Section 2.5. To make this clear, we have added the following statement at L260:

*Here and in the following, BrO VCD and aerosol optical depth (AOD) refer to tropospheric columns obtained from the MAX-DOAS measurements with their specific sensitivity as quantified by the corresponding averaging kernels.*

**P18, L364: By the different meteorological conditions and also by the location of each research site.**

We are not sure what the reviewer means with this comment.

**P19, L390: Please, state the meaning of having higher bromide-to-sodium ratio.**

We have extended the corresponding sentence as follows:

*The molar bromide to sodium ratio in aerosols shown in the bottom panel of Figure 8 is strongly enhanced compared to mean sea water with a ratio of only 1.8e-3, which means that an enrichment of the aerosol in bromide occurred either prior to the dispersion of the aerosol particles or by scavenging of gaseous bromide at the particle surfaces*

**P19, L401: How does the surface BrO daily evolution compare to the work of Nasse et al. 2019 where, with a LP-DOAS, they detected up to 60ppt around noon?**

The seasonal mean profiles shown here cannot be directly compared to the case studies discussed in Nasse 2019. Furthermore, as discussed by Nasse (2019), the MAX-DOAS instrument averages over a certain height altitude, whereas the LP-DOAS directly observes the concentration near the ground, which means that MAX-DOAS surface concentrations are lower than from LP-DOAS when the BrO profile is restricted to a shallow altitude. However, the diurnal variability is qualitatively in agreement, with both instruments showing a peak in the morning and a secondary peak prior to sunset. We would like to refer to the thesis of Nasse (2019) for a more detailed comparison between MAX-DOAS and LP-DOAS at Neumayer.

**P21, L448: Regarding BrO getting into the free troposphere, what is the evolution of the boundary layer height?**

Unfortunately, there are no continuous measurements of the boundary layer height available. At NM, ceilometer measurements are available for a limited period of time, but these have not yet been analysed. It can, however, be expected the boundary layer is more shallow during the night than during the day, and the boundary layer height is probably largest in summer when solar insulation is at its maximum.

**P22, figure 10: Is this figure considering only BrO data above detection limit?**

As stated in L441, only BrO VCDs > 1e13 molec/cm^2 are included in this analysis since otherwise the Gaussian and exponential fit would not yield reliable result. This value is about three times the VCD detection limit.

**P22, L56-457: p-values close to 0 indicate the significance of the R obtained.**

To our knowledge, the p-value quantifies the probability that the zero-hypothesis is fulfilled, which is in this case the hypothesis that two quantities are uncorrelated, and does not provide any measure for the significance of the Person's correlation coefficient

**P23, figure 11: Which BrO and aerosol from MAX-DOAS are used? Surface values? (also in P24, L:461)**

This is explained in the caption of Figure 11, as well as in Section 2.7 (in particular Tables 4 and 5). In Figure 11, columnar and surface values are labelled in different colours as explained in the caption.

**P25, L523: The source-receptor analysis seems very clarifying. Please detailed a bit better the procedure behind this sort of analysis (e.g., which altitude of the BrO profile is used for this analysis)?**

This is explained in detail in Section 2.6. The source-receptor analysis relates the residence time of air parcels is performed including the BrO VMR and aerosol extinction at all altitude layers. To make this clearer, the sentence in L291 has been modified as follows:

"First, the BrO VMR and aerosol extinction observed at the observation site *including measurements at all altitude layers* was assigned to the respective air parcel of the back-trajectory simulations."

**P25, L527: Do observations at Belgrano or Halley support the FRIS as a BrO hot spot? Also P26, L542; P27, L547.**

We have added the following statement to L529:

*Observations of the BrO VCD at Belgrano station at the southern end of the Weddell Sea do not exceed 2e13 molec/cm² (Prados-Roman et al., 2018), but the measurement period does not start before early October at this location far south, and the springtime BrO season is already at its end during this time at NM (see Figure 7). Longpath-DOAS measurements at Halley Bay show maximum BrO VMR of about 20 ppt during spring (Saiz-Lopez et al., 2007), but these were located at the eastern coast of the Weddell Sea where our source-receptor analysis indicates a strong spatial gradient between on-shore and off-shore BrO.*

**P27, L557-558: Do observations at Dumont d'Urville support the TNB as an aerosol hot spot?**

We are not aware of any continuous aerosol extinction measurements at Dumont d'Urville. However, the point in our study is that extinction is high when air coming from this region reaches Arrival Heights due to particular meteorological conditions, and not that aerosols are enriched in Terra Nova Bay region in general.

**P30, figure 17: The time referred to is 00:30. Given the high SZA, how can the stratospheric influence be excluded?**

00:30 UTC is around noon at Arrival Heights.

**P30, L586-588: Can a stratospheric intrusion be ruled out? Are O3 values available?**

The following sentence has been added at L584:

*An enhancement of BrO owing to a stratospheric intrusion can be ruled out since the altitude of the air parcels was not higher than 2.6 km, and the ozone VMR observed at AH did not exceed 20 ppt.*

**P32, L610: Please provide the BrO VCD also in ppt (easier to compare with other ground-based observations).**

The sentence in L610 compares the BrO VCD detection limit from satellite and ground-based measurements. It is not clear to us how a VCD (unit molec/cm^2) can be expressed as mixing ratio in units of ppt.

**P32, L613-615: (Snow surface as a BrO sink during the day). This may contradict with e.g. the work of Saiz-Lopez et al. 2007 with a LP-DOAS. Same applies with the observed diurnal variation at NM with an early morning peak.**

Sorry, this was a typo. The sentence should read "*The reduced BrO concentrations observed near the ground during NDJ and FMA…*", not "*…ASO and NDJ…*". The model study of Saiz-Lopez et al. (2007) refers to BrO at spring time.

**Technical Corrections:**
**P1, L1: Polar Regions shall be polar regions (no block letters).**
Done.

**P5, L109: Katabatic shall katabatic (no block letter).**
We did not find any occurrence of the term "katabatic" with the first letter in capital, except at the beginning of a sentence.

**P15, L319: 28. August shall be 28th August. This happens to all dates throughout the manuscript.**
We have revised date and time values according to the Copernicus standard in the revised manuscript.

**P31, L591: A ", " is missing after Thus**
Done

**P31, L598: The "dynamics chemistry"? do the authors refer to the chemistry and the dynamics?**
This should read "*dynamics and chemistry*" and has been corrected.

**REPLIES TO ANONYMOUS REFEREE #2**

**The study by Friess et al uses two decades of tropospheric BrO measurements in Antarctica, performed at two sites, to investigates the links between trop. BrO and other geophysical quantities, either measured or modelled. From this and the temporal variability of trop. BrO, the authors draw conclusions on the source regions and formation mechanisms of this species. The authors also indentify an interesting event of BrO transport across the continent. The paper presents a unique database which is seriously analyzed, it is also well structured and perfectly in the scope of ACP so this work should be published.**

**On the content, my main remark is that on several occasions (e.g. L 326, L.485 and L592, see below), when interpreting measurements with models, the authors assume the models more realistic and this is not discussed. If they are reasons to think the models are good enough in Antarctica, this is an interesting info to add in the paper since one may expect them to be limited by the available sampling there. If not, maybe a few words of caution about the models could be added in the interpretation.**

**Other than that, my remarks mainly aim to improve the readability.**

**P5. Table 1 should also include AH and NM.**
AH and NM were added to Table 1

**P7 L180 The sentence starting with 'DSCDs' is complicated and misleading (it describes SCDs not dSCDs after the i.e.), I suggest to break it in two and put the subjects closer to the verbs.**
This sentence has been replaced by
*Differential slant column densities (dSCDs) of BrO and the oxygen collision complex $O_4$ were determined from scattered light spectra using the DOASIS software developed at IUP Heidelberg (Kraus, 2006). These represent the integrated trace gas concentrations along the light path of a trace gas relative to a reference spectrum, usually measured in zenith sky. $O_4$ dSCDs serve as a proxy for the light path through the atmosphere and thus for the abundance of aerosols.*

**P8 L193 'non linear constant intensity offset'. I think the authors mean that this is a non linear term in the DOAS fits because it is additive on the intensity. If so 'Non-linear constant' reads weird. Maybe replace 'non-linear' by 'additive'? Or explain more otherwise.**
This sentence has been replaced by:
*In order to account for possible instrumental stray light, a polynomial of zeroth order is fitted to the intensity spectrum.*

**P8 L206. The sentence is long and a bit messy. Consider dropping '(which are ....)' or breaking in two.**

The statement in brackets *"(which are later converted to BrO VMR profiles)"* has been removed from the sentence

**P8 L 209. 'Most probable atmospheric state'. This reads an overstatement, even if the sentences after bring some clarifications. An OEM algorithm maximizes a metrics, but there is subjectivity involved in many ways (gridding, forward model, covariance matrix, convergence criterion etc...), not talking about the metrics itself. I would just remove 'most probable' unless the authors want to develop.**

Optimal estimation yields the most probable state in a mathematical sense (given a normal distribution of the probabilities), but of course many choices, such as the vertical grid and an arbitrary choice of the a priori constraints, have a significant impact on the solution. We have therefore replaced *"the algorithm determines the most probable atmospheric state"* with *"the algorithm determines a maximum a posteriori solution for the atmospheric state"*.

**P 9 Table 3 should include the surface albedo.**

The surface albedo (0.98) has been added to Table 3.

**Using layers of different thickness in the retrieval is interesting. I do not follow however why the small altitude difference explains that it does not work for AH. Did the authors do some tests to check this statement? If so explain, if not, reformulate the very affirmative 'most probably', as it could come also from different S/N in the DSCDs, or from the different elevation sequences.**

We have made many attempts to find settings that yield a stable retrieval for AH in case of a non-equidistant grid. Unfortunately, the reasons why this was not successful are not clear, but we believe that this is not because of different S/N or different elevation sequences, because the retrieval behaves in a robust manner when applying a regular grid.

*Unfortunately, the application of a non-regular grid led to an unstable behaviour for the retrieval of vertical profiles at AH despite several attempts to find appropriate settings. It is not clear why this instability occurs, but a possible reason for this....*

**Figure 2: please add the corresponding date so that we can relate the AKs to a given angular sequence.**

Date and time of the AVKs were added to the caption of Figure 2.

**It is mentionned that the AK are for 'clear-sky', but what about the aerosol load?**

The aerosol load was really small, with AODs of 2e-3 at NM and 2.7e-2 at AH. This information has been added to the caption of Figure 2.

**I ask that since I find weird the minimum sensitivity at the altitude of observations for AH. If the authors can further develop this last point, it s worth.**

It is not really clear to us why there is a minimum in sensitivity for the layer where the instrument resides. The instrument is, however, at located the top of this layer, and the fact that the sensitivity is reduced for air masses below the instrument may explain this finding.

**p 11 L.265 et seq. 'This illustrates that obs ...' -> I do not follow here. Maybe I miss something, but the described improvement in dofs is for adding 1° elevation above the horizon, right? And the scope of the statement seems very large, but at NM, you have special conditions with the high surface albedo. It seems to me that you should describe the effect on the dofs of removing the negative angles, and give arguments for the fact that this is also true at low albedo (e.g. refering to other studies in warmer areas).**

What we intended to state here is the following: (1) Including the 1° elevation angle at NM improves the information content; (2) DOFS for aerosol retrieval is higher at NM than at AH; (3) DOFS for BrO is higher at AH than at NM, showing that adding downward viewing angles for elevated sites enhances the information content for trace gases. We have successfully applied this to elevated mid-latitude sites, but we do not want to expand this topic too much in the manuscript since the focus of our study is on physics and chemistry, not on measurement techniques.

**p 11 L.274 and very often later in the paper: there is a dot after the day number in the date, please**
See our reply to your next comment.

**Figure 3: Please add 'HYSPLITT' at the begining of the caption. The date, here and often after in the paper, does not follow copernicus standards, and add UTC or LT for the time.**
**remove**
We have revised date and time values according to the Copernicus standard in the revised manuscript, and added UTC where appropriate.

**Figure 4: legend and axis text should be larger**
The font size in Figure 4 has been increased. Also, in response to Reviewer #1, the error of the BrO profile has been added.

**p 15 L.318 'Figure' should be 'Fig.' (here and across the paper) except at the beginning of a sentence, according to Copernicus standards.**
We have replaced "*Figure*" with "*Fig.*" throughout the document according to the Copernicus standards.

**L.326 'The difference ... can be attributed to the lower sensitivity of the MAX ..." There arealso errors in the back trajectory modeling. Is there a reason why the authors do notconsider it here?**
There is of course a substantial uncertainty in the modelled trajectories due to a low number of meteorological observations in and around Antarctica. The sentence has been modified as follows:
*The difference between the shape of the sea ice contact time profile and the BrO profile can be attributed both to uncertainties in the back trajectories and to the lower sensitivity of MAX-DOAS to higher altitudes…*

**'no BrO is detected above the detection limit' -> 'BrO remains under the detection limit'?**
Following a comment by Reviewer #1, this sentence has been modified as follows:
*Afterwards, the air masses above NM were not in contact with sea ice anymore and only very small BrO VMR below 1 ppt were detected…*

**L 344. I would add ', in this place, ' before the second BrO.**
Done.

**In Fig. 7 et seq. (or in the caption) I would indicate the studied periods for the two sites.**
We have added "*based on all available data*" to the caption, while the time ranges for both instruments are listed in Table 3.

**L. 485 et seq., again, the model error for the wind speed over sea ice is not considered. It seems safe to assume that a measured wind speed is more accurate than a modelled one, especially in Antarctica. Unless I missed something, maybe the authors could add a few words on that?**
This paragraph discusses the correlation between wind speed and BrO. It turns out that BrO and wind speed measured at the observation site are positively correlated, but BrO and wind speed modelled along the trajectories are (slightly) negatively correlated. There is of course a significant uncertainty in the modelled wind speed, but there would be something seriously wrong with the meteorological model if this would lead to a change in sign of the correlation coefficient.

**L. 499 to 502. Break the sentence in two.**
This sentence has been replaced by the following:
*This could be caused by BrO release directly from sea spray, as observed in mid-latitudes and the tropics where BrO was found to be present in the lower ppt range at coastal stations and during ship cruises (e.g., Leser et al., 2003, Saiz-Lopez et al., 2004). Another explanation could be the release of reactive bromine from the snow surface after deposition of saline particles, which is a likely source for the morning time peak in BrO during FMA at NM discussed in Section 3.2.*

**P 26 L 534-535 'This area' -> Seems to refer to 'the coastline east of NM' 4 lines before. I think this would help the readability to be more explicit e.g. 'this coast'**
We have replaced "…*if the air comes from this area…*" with "…*if the air travels along this coastline…*".

**Across the whole Section 3.4, it would help the reader to have the important places for each site on the maps of fig 12 13 14 15. One of the suplots of fig 12 and 14 could pinpoint the important areas for NM, the same for AH on fig 13 and 15 (e.g. with symbols described in the caption)**
This is technically difficult because the label font would be hard to read on top of the coloured maps. This is why we chose to introduce Fig. 1 and Table 1 that show the geographical names and explains the abbreviations used in the manuscript.

**L 592 again, the model seems implicitly assumed more realistic that the measurements.**
It is not clear to us whether the referee refers to the photochemical model calculations from other authors we refer to in this paragraph, or to the back trajectory calculations.

**L 619: 'am and pm' -> 'morning and afternoon' ?**
"*am and pm*" has been replaced by "*morning and evening*"

**- L 637 missing 'from' after released? the whole sentence reads tautological: reactive bromine serves as a source of reactive bromine?**
This sentence has been replaced by the following:
*This indicates that the open ocean serves as a source for reactive bromine, with the bromine being released from…*

**Minor comments:**

**-the block letters in the title does not seem all apropriate**
The capitalisation has been changed in the title as follows:
"*Source mechanisms and transport patterns of tropospheric BrO: Findings from long-term MAX-DOAS measurements at two Antarctic stations*"

**-in the author list, it reads weird that the affiliation of the 2nd author is the 3rd one**
Thank you for spotting this mistake, the author affiliations are now numbered in ascending order.

**-L 349 'is' -> 'ice'**
Done.

**-L 426 'indcates'-> 'indicates'**
Done.

**-L 473 comma missing after 'There'?**
Replaced "*There open ocean*" with "*The open ocean*".

**-L 515 Sander et al should be between parenthesis**

Done.

**-L 582 'resides' -> 'resided'**
Done.